**Enviro-HIRLAM online integrated meteorology-chemistry modelling system: strategy, methodology, developments and applications (v. 7.2)**

*Alexander Baklanov (1, a), Ulrik Smith Korsholm (1), Roman Nuterman (2), Alexander Mahura (1, b), Kristian Pagh Nielsen (1), Bent Hansen Sass (1), Alix Rasmussen (1), Ashraf Zakey (1, c), Eigil Kaas (2), Alexander Kurganskiy (2,3), Brian Sørensen (2), Iratxe González-Aparicio (4)*

(1) Danish Meteorological Institute (DMI), Copenhagen, Denmark; (2) Niels Bohr Institute, University of Copenhagen, Denmark; (3) Russian State Hydrometeorological University, St.Petersburg, Russia; (4) European Commission, DG - Joint Research Centre, Institute for Energy and Transport, The Netherlands; (a) now at: World Meteorological Organization (WMO), Geneva, Switzerland; (b) now at: University of Helsinki, Finland; (c) now at: The Egyptian Meteorological Authority, Cairo, Egypt.

*Corresponding author email: abaklanov@wmo.int; alb@dmi.dk*

**Abstract:** The Environment – HIgh Resolution Limited Area Model (Enviro-HIRLAM) is developed as a fully online integrated numerical weather prediction (NWP) and atmospheric chemical transport (ACT) model for research and forecasting of joint meteorological, chemical and biological weather. The integrated modelling system is developed by DMI in collaboration with several European universities. It is the baseline system in the HIRLAM Chemical Branch and used in several countries and different applications. The development was initiated at DMI more than 15 years ago. The model is based on the HIRLAM NWP model with online integrated pollutant transport and dispersion, chemistry, aerosol dynamics, deposition and atmospheric composition feedbacks. To make the model suitable for chemical weather forecasting in urban areas the meteorological part was improved by implementation of urban parameterizations. The dynamical core was improved by implementing a locally mass conserving semi-Lagrangian numerical advection scheme, which improves forecast accuracy and model performance. The current version 7.2, in comparison with previous versions, has a more advanced and cost-efficient chemistry, aerosol multi-compound approach, aerosol feedbacks (direct and semi-direct) on radiation and (first and second indirect effects) on cloud microphysics. Since 2004 the Enviro-HIRLAM is used for different studies, including operational pollen forecasting for Denmark since 2009, and operational forecasting atmospheric composition with downscaling for China since 2017. Following main research and development strategy the further model developments will be extended towards the new NWP platform - HARMONIE. Different aspects of online coupling methodology, research strategy and possible applications of the modelling system, and 'fit-for-purpose' model configurations for the meteorological and air quality communities are discussed.

## 1. Introduction

During the last decades a new field of atmospheric modelling - the chemical weather forecasting (CWF) - is quickly developing and growing. However, in most of the current studies this field is still considered in a simplified concept of the off-line running of atmospheric chemical transport (ACT) models with operational numerical weather prediction (NWP) data as a driver (Lawrence at al., 2005). A new concept and methodology considering the "chemical weather" as two-way interacting nonlinear meteorological and chemical/aerosol dynamics processes of the atmosphere have been recently suggested (Grell et al., 2005; Baklanov and Korsholm, 2008; Baklanov, 2010; Grell and Baklanov, 2011). First attempts at building online coupled meteorology and air pollution models for environmental applications were done in the 1980s, cf. Baklanov (1988), Schlünzen and Pahl (1992), Jacobson (1994). For climate applications the first coupled chemistry-climate models were developed and used in the 1990s, cf. Jacobson (1999, 2002), de Grandpré et al. (2000), Steil et al. (2003), Austin and Butchart (2003). More detailed overview of the history and current experience in the online integrated meteorology-chemistry modelling, importance of different chains of feedback mechanisms for meteorological and atmospheric composition processes are discussed for USA (Zhang, 2008) and European (Baklanov et al., 2014) models. Klein et al. (2012) extended applications of coupled models for "biological weather", defined as "the short-term state and variation of concentrations of bioaerosols", in particular for pollen modelling and forecasting.

The online integration of meso-meteorological models (MetM) and atmospheric aerosols and ACT models gives a possibility to utilize all meteorological 3D fields in the ACT model at each time step and to consider nonlinear feedbacks of air pollution (e.g. atmospheric aerosols) on meteorological processes / climate forcing and then on the chemical composition of the atmosphere. This very promising way for future atmospheric modelling systems (as a part of and a step toward the Earth System Modelling, ESM) will lead to a new generation of seamless coupled models for meteorological, chemical and biochemical weather forecasting. Seamless approach for 'one atmosphere' integrated meteorology-chemistry/aerosols forecasting systems is analysed by the COST Action ES1004 EuMetChem (see e.g. Baklanov et al., 2015) and overview of the current state of online coupled chemistry-meteorology models and needs for further developments were published in (Zhang, 2008; WMO, 2016; Baklanov et al., 2017; Sokhi et al. 2017).

The methodology on how to realize the suggested integrated concept was demonstrated on an European example of the Enviro-HIRLAM (Environment – High Resolution Limited Area Model) integrated modelling system (Baklanov et al., 2008a; Korsholm, 2009). Experience from first HIRLAM community attempts to include pollutants into the NWP model (Ekman, 2000) and from pioneering online coupled meteorology-pollution model developments of the Novosibirsk science

school (Marchuk, 1986; Penenko and Aloyan, 1985; Baklanov, 1988) was actively used for developments of the Enviro-HIRLAM modelling system.

The Enviro-HIRLAM is developed as a fully online integrated NWP and ACT modelling system for research and forecasting of meteorological, chemical and biological weather. The integrated modelling system is developed by DMI and other collaborators (Chenevez et al., 2004; Baklanov et al., 2008a, 2011b; Korsholm et al., 2008, 2009; Korsholm, 2009) and included as the baseline system of the Chemical Branch of the HIRLAM consortium (Figure 1).

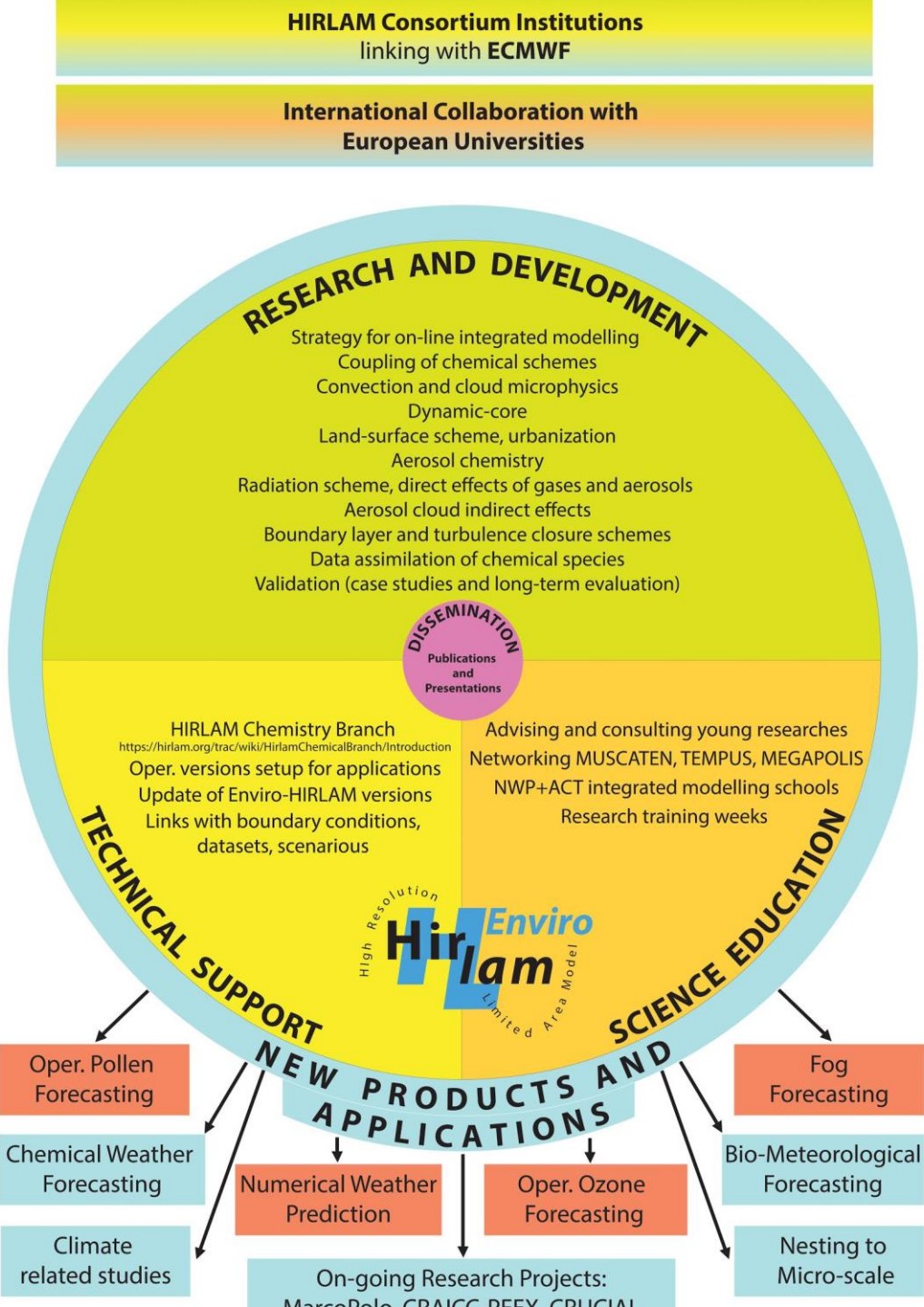

**Figure 1.** General scheme of international collaboration, research and development, technical support and science education for the online integrated Enviro-HIRLAM: 'Environment – HIgh Resolution Limited Area Model'.

The model development was initiated at DMI more than 15 years ago and it is used now in several countries. The modelling system is being used for different completed and ongoing research projects (FP6 FUMAPEX; FP7 MEGAPOLI, PEGASOS, MACC, TRANSPHORM, MarcoPolo; NordForsk NetFAM, MUSCATEN, CarboNord, CRAICC-PEEX, CRUCIAL; COST Actions – 728, 732, ES0602 ENCWF, ES1004 EuMetChem), and for operational pollen forecasting in Denmark since 2009 (Rasmussen et al., 2006; Mahura et al., 2006b) and operational atmospheric composition (with focus on aerosols) for China since Nov 2016 (Mahura et al., 2016; 2017). Following main strategic plans (Baklanov, 2008; Baklanov et al., 2011a) within HIRLAM-B,-C projects further developments of the modelling system will be shifting to new NWP platform (from HIRLAM to HARMONIE) and a close collaboration with the ALADIN (Aire Limitée Adaptation dynamique Développement InterNational) community was initiated in 2014.

In this paper an overall description of the current version of the Enviro-HIRLAM coupled modelling system with improved parameterisations of meteorology-composition two-way interactions, main steps in its development and examples in different application areas for air quality, weather and pollen forecasting are considered for the first time. Section 2 provides a detailed description of the Enviro-HIRLAM modelling system and its key developments in the meteorological core, chemistry and aerosol dynamics parts, aerosol-meteorology interactions, models urbanisation and improvements of numerical algorithms. Section 3 describes a few types of Enviro-HIRLAM applications for meteorological and environmental forecasting and assessment studies. Sections 4 and 5 continue discussions and summarise the model applicability and provide recommendations for future research. Annex 1 includes brief information about the Enviro-HIRLAM model development history. A list of acronyms is provided in Annex 2.

## 2. Enviro-HIRLAM modelling system description

### 2.1. Modelling system structure

The Enviro-HIRLAM is a fully online coupled (integrated) NWP and ACT modelling system for research and forecasting of meteorological, chemical and biological weather (see schematics in Figure 2). The modelling system was originally developed by DMI and further with other collaborators, and now it is included by the European HIRLAM consortium as a baseline system in the HIRLAM Chemical Branch (http://hirlam.org/index.php/documentation/chemistry-branch). It was the first meso-scale online coupled model in Europe that considered two-way indirect feedbacks between meteorology and chemistry/aerosols (WMO-COST, 2008).
The following main steps of the model development were realised such as: (i) model nesting for high resolutions, (ii) improved resolving PBL and surface layer structure, (iii) urbanisation of the NWP model, (iv) improvement of advection schemes, (v) emission inventories and models, (vi) implementation of gas-phase chemistry mechanisms, (vii) implementation of aerosol dynamics, (viii) realisation of aerosol feedback mechanisms.

The first version was based on the DMI-HIRLAM NWP model with online integrated pollutant transport and dispersion (Chenevez et al., 2004), chemistry, deposition and indirect effects (Korsholm et al., 2008; Korsholm, 2009) and later aerosol (only for sulphur particles) dynamics (Baklanov, 2003; Gross and Baklanov, 2004). To make the model suitable for chemical weather forecasting in urban areas the meteorological part was improved by implementation of urban sub-layer parametrisations (Baklanov et al., 2008b; Mahura et al., 2008a; González-Aparicio et al., 2013). The model's dynamic core was improved by adding a locally mass conserving semi-Lagrangian numerical advection scheme (Kaas, 2008; Sørensen, 2012; Sørensen et al., 2013), which improves forecast accuracy and enables performing longer runs. More details of the system development history is presented in the Annex 1.
The current version of Enviro-HIRLAM (Nuterman et al., 2013; Nuterman et al., 2015) is based on the reference HIRLAM v7.2 with a more advanced and effective chemistry scheme, multi-compound modal approach aerosol dynamics modules, aerosol feedbacks on radiation (direct and semi-direct effects) and on cloud microphysics (first and second indirect effects). This version is continuously under development and evaluation for various weather and air quality related applications (in particular, within the COST Action ES1004 where the above mentioned effects were extensively discussed, see, e.g., in Baklanov et al. 2014).

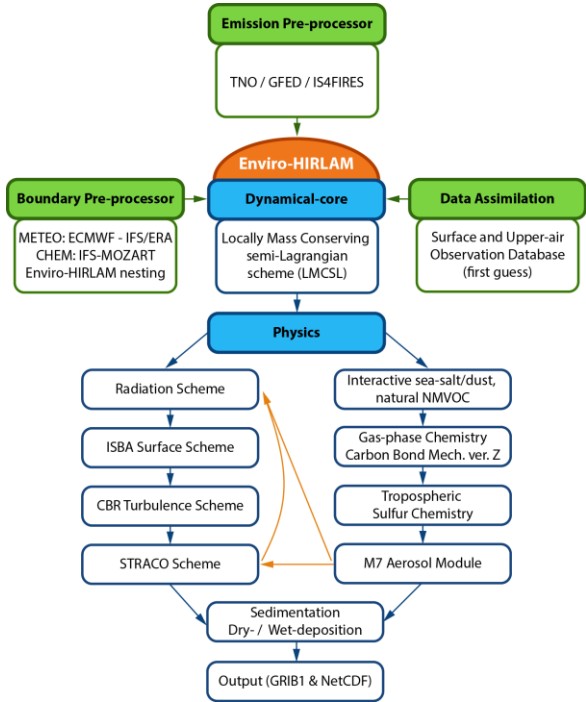

Figure 2: Schematics of the Enviro-HIRLAM modelling system.

Vertical and horizontal resolutions of the model are flexible. Limitations, e.g. due to the hydrostatic approximation, are provided (min 1,5 km of the horizontal resolution for flat terrains, e.g. for Copenhagen).

### 2.2. Meteorological core of the system

The first version of Enviro-HIRLAM was based on a previous version HIRLAM-tracer and at its meteorological core lies DMI-HIRLAM, version 6.3.7 employed for limited area short range operational weather forecasting at DMI (Chenevez et al., 2004). The current model version used in studies is based on the reference version of the HIRLAM community meteorological NWP model HIRLAM version 7.2 and online-coupled environmental block (so-called, the Enviro-) allowing to take into account spatial-temporal evolution of atmospheric chemical and biological aerosols driven by meteorology from NWP block.

HIRLAM is a hydrostatic NWP model which is used for both research and operational purposes. The model provides forecast of main meteorological fields, including: air temperature and specific humidity, atmospheric pressure, wind speed and direction, cloud cover and turbulent kinetic energy (TKE) based on forward in time integration of the primitive equations (dynamical core) (Holton, 2004) and physical processes such as radiation, vertical diffusion, convection, condensation, etc. (physical core).

The detailed NWP HIRLAM description can be found in the HIRLAM reference guide science documentation (Undén et al., 2002) and its following upgrades and modifications (see more details at http://www.hirlam.org).

The hydrostatic approximation of the model can be a limitation to increase the resolution for urban simulations. However, sensitivity tests for a medium size city demonstrated that the 2.5 km was the optimal resolution, allowing at the same time to obtain satisfactory reproducibility of the large scale processes and to explore the urban effects at local scale (González-Aparicio et al. 2013). For other metropolitan areas such as Paris, Rotterdam, St. Petersburg, Shanghai - a similar resolution was chosen, whereas for Copenhagen (with its flat terrain) the highest suitable resolution tested was 1.5 km and provided reasonable verification results (Mahura et al., 2006a, 2008bc, 2016). Within a selected metropolitan area there could be only a few grid cells having 100% representation of the urban fraction, but taking into account all urban grid cells, the boundaries of the cities (number of cells) could be substantially larger. Moreover, most of existing parameterizations in the physics core of any NWP model might need a revision when resolutions of 1 km and finer are used.

Following the main strategic development within HIRLAM (HIRLAM-B and -C projects), there are plans for further developments of Enviro-HIRLAM shifting to new non-hydrostatic NWP platform (e.g. HARMONIE model) and incorporating chemistry modules and aerosol–radiation–cloud interactions into the future integrated system (Baklanov, 2008; Baklanov et al., 2011a).

The new non-hydrostatic version under HARMONIE is under development and only some elements are realised so far. The non-hydrostatic HARMONIE-AROME model includes only some aerosol effects. The physics included in this version of HARMONIE has recently been detailed by Bengtsson et al. (2017). HARMONIE-AROME is based partly on Meso-NH (Mesoscale Non-Hydrostatic atmospheric model), which is a cloud resolving model that includes state-of-the-art chemistry and aerosol interactions (e.g. Berger et al. 2016). However, Meso-NH cannot be run as a near real time NWP model, as it is possible with Enviro-HIRLAM.

*2.3. Atmospheric chemistry*

a) Tropospheric Sulfur Cycle

The simple tropospheric sulphur cycle chemistry module in Enviro-HIRLAM, used for long-term runs (up to one year), is based on the sulfur cycle mechanism developed by Feichter et al. (1996) treating three prognostic species dimethyl sulfide (DMS), sulfur dioxide ($SO_2$) and sulfate ($SO_4^{2-}$). The mechanism includes DMS and $SO_2$ oxidation by hydroxyl (OH) and DMS reactions with nitrate radicals ($NO_3$) in the gas-phase part. The heterogeneous aqueous phases chemistry comprises of $SO_2$ oxidation reactions by $H_2O_2$ and $O_3$. Accounting for dissolution effects of $SO_2$ in the aqueous phase is performed according to Henry's law. An output of global chemistry transport model MOZART (Horowitz et al., 2003) is used to prescribe three dimensional oxidant fields of OH, $H_2O_2$, $NO_2$, and $O_3$.

The sulfate produced in the gas-phase is referred to the gases and can be condensed on pre-existing aerosols or to nucleate by the aerosol microphysics M7 module (see Sect. 2.4). Moreover, in-cloud produced sulfate is accumulated on the pre-existing accumulation and coarse mode aerosols.

The tropospheric sulfur cycle chemistry is used together with M7 aerosol microphysics module because of its relative simplicity and low computational cost. The CBM-Z gas-phase chemistry (see the next section) is not interfaced with the M7 aerosol module because of several reasons: 1) the aerosol microphysics module does not include Secondary Organic Aerosols, therefore, there is no need of complex gas-phase mechanism with Volatile Organic Compounds related reactions and 2) it is too computationally expensive to use CBM-Z together with M7 for both weather and atmospheric composition prediction.

b) Gas-phase chemistry

The gas-phase chemistry scheme consists of sets of chemical schemes running from simple schemes for Chemical Weather Forecasts to highly complex schemes for research and case studies. In order to make the model computationally efficient for different applications and operational forecasting several condensed atmospheric chemical schemes have been tested into Enviro-HIRMAM since the first version of the model system was realised (Korsholm, 2009; Gross and Baklanov 2004). In the current version of Enviro-HIRLAM the tropospheric condensed Carbon–Bond Mechanism version Z (CBM–Z) (Zaveri and Peters, 1999), a variant of CBM–IV gas-phase chemistry scheme (Gery et al., 1989), with a fast solver based on radical balances (Sandu et al., 2006) has been implemented in the model. CBM-Z uses lumped species that represent broad categories of organics based on carbon bond structure. It is closely related to CBM-IV which is widely used in air pollution evaluations, but with expansions to include reactions that are important in the remote troposphere. It also uses the most general organic category (PAR for paraffin) to represent miscellaneous carbon content so that carbon mass is conserved.

Six environmental/smog chamber experiments were used to validate the gas-phase schemes as box models and within a regional climate model (Shalaby, 2012; Shalaby et al., 2012). The Tennessee Valley Authority (TVA) and the EPA chamber experiments were used to evaluate the different gas-phase schemes and different chemical solvers. Namely, TVA005 and TVA006 are designed to test the simple system of NOx; TVA068 is designed to test a simple mixture of VOC with very high NOx. EPA069A, EPA073A and EPA150A are used to validate the schemes with low NOx concentration and high VOC concentration.

c) Chemical Solvers

Calculating the time evolution of gas-phase chemistry requires a numerical integration of a set of stiff ordinary differential equations (ODE) and is among the most computationally expensive operations performed in a photochemical grid model. The equations for photochemical production and loss are computationally expensive because they form a stiff numerical system. The photochemical mechanisms described above were implemented using two different chemical solvers to solve the tendency equation for photochemical production and loss: (1) the Rosenbrock (ROS) solver (Sandu et al., 1997 and Hairer and Wanner, 1996) as implemented by the Kinetic Preprocessor (KPP) (Sandu et al., 2006); and (2) the computationally rapid radical balance method (RBM) of (Sillman, 1991). RBM utilizes the fact that much of the complexity of tropospheric chemistry stems from the $HO_x$ radical family (OH, $HO_2$ and $RO_2$), which has a limited set of sources and sinks. The method solves reverse-Euler equations for OH and $HO_2$ based on the balance between sources, sinks and (if applicable) prior concentrations at the start of the time step. Reverse Euler equations for other species are solved in a reactant-to-product order, in some cases involving pairs of rapidly interacting species, and with some modifications to increase accuracy in exponential decay situations. The procedure is equivalent to a reverse Euler solution using sparse-matrix techniques, but with the matrix inversion linked specifically to the behaviour of OH and other species in the troposphere. Prior work tested several atmospheric chemistry mechanisms in the model taken into account different chemical solvers, we select the photochemical mechanism CBM-Z because it affords a reasonable trade-off between accuracy and computational efficiency. During the prior work including the validation stages of the gas-phase schemes (results not shown) we used KPP to generate the Fortran code of three different gas-phase schemes CBM-Z (Zaveri and Peters, 1999), GEOS-CHEM (Evans et al. 2003) and the Regional Atmospheric Chemistry Model "RACM" (Stockwell et al,1997). In order to fit within our main aim of the chemical weather predication, we didn't use both of GEOS-CHEM and RACM because they are very computationally expensive schemes due to their extensive number of chemical reactions. The KPP provides a flexible tool to generate a well coded chemical mechanism according to the user choice of a given ODE solver. We use KPP tools to create the gas-phase chemical mechanisms including the solvers for three chemical mechanisms. Usually, the Rosenbrock solver is selected for most of simulations due to its ability as a fast computational solver (Sandu et al 1997).

### d) Photolysis Rates

Photolysis rates are determined as a function of various meteorological and conditional inputs. Rates for specific conditions are determined by interpolating from an array of pre-determined values. The latter is based on the Tropospheric Ultraviolet-Visible Model (TUV) developed by Madronich and Flocke (1999), using a pseudo-spherical discrete ordinates method (Stamnes et al., 1988) with 8 streams. The 8-stream TUV is the most accurate method for determining photolysis rates but is computationally too expensive for use in 3D models. Photolysis rate constants are calculated using the Fast-J radiative transfer model (Wild et al., 2000) with O(1D) quantum yields updated to JPL2003 (Sander et al., 2003).

For simplicity, photolysis rates are estimated as the following. At first, for the simple reactions the photolysis rates are estimated as a function of number of parameters such as meteorological and chemical inputs including altitude, solar zenith angle, overhead column densities for $O_3$, $SO_2$ and $NO_2$, surface albedo, aerosol optical depth, aerosol single scattering albedo, cloud optical depth and cloud altitude. At second, for the complex reactions, the photolysis rates are estimated as lookup table using the TUV model. TUV is run offline and calculated a lookup table of the photolysis rates, and then this lookup table is implemented under different weather conditions inside the model.

Photolysis rates can be significantly affected by the presence of clouds. Cloud optical depths are determined using the random overlap treatment described by Feng et al. (2004), which assumes that cloudy and cloud-free sub-regions in each model grid box randomly overlap with cloudy and cloud-free sub-regions in grid boxes located above or below (Briegleb, 1992). The method used to correct for cloud cover is based on Chang et al. (1987), which requires information on cloud optical depth for each model grid cell. Optical depth is used to reduce photolysis rates for layers within or below clouds to account for UV attenuation or to increase photolysis rates due to above-cloud scattering. In general, below cloud photolysis rates will be lower than the clear sky value due to the reduced transmission of radiation through the cloud. Similarly, photolysis rates are enhanced above the cloud due to the reflected radiation from the cloud. Cloud optical depths and cloud altitudes from Enviro-HIRLAM are used in the photolysis calculations, thereby directly coupling the photolysis rates and chemical reactions to meteorological conditions at each model time step.

### e) Heterogeneous chemistry

Many gas-phase species are water soluble and sulphate and ammonia together with water take part in binary/ternary nucleation. In order to consider these processes, a simplified liquid-phase equilibrium mechanism with the most basic equilibria is included in NWP-Chem-Liquid. The "NWP-Chem-Liquid" is a thermodynamic equilibrium model, described in Korsholm et al. (2008). This equilibrium module is solved using the analytical equilibrium iteration method (Jacobson, 1999). The reactions are summarized in Korsholm (2009) and the module will be updated to include the impact of organic compounds from anthropogenic and biogenic sources.

## 2.4. Aerosol formation, dynamics and deposition

### a) Aerosol dynamics module

The first aerosol module in Enviro-HIRLAM was based on the CAC (Chemistry-Aerosol-Cloud) model with the modal approach for description of aerosol size distribution (Baklanov, 2003; Gross and Baklanov, 2004) and considered only sulfur-type aerosols (Korsholm, 2009).

The current version of the Enviro-HIRLAM model has M7 aerosol microphysics module (Vignati et al., 2004) together with aerosol removal processes ported from ECHAM5-HAM climate model (Stier et al., 2005). There are two types of particles considered: insoluble and mixed (water-soluble) particles. The particles are split into seven classes depending on particle size and solubility by means of "pseudomodal" approach. Four classes are used to represent mixed particles, i.e., nucleation, Aitken, accumulation, and coarse modes, and another three classes are for the insoluble (Aitken, accumulation, and coarse modes). Four predominant aerosol types are included - black carbon (BC) and primary organic carbon (OC), sulfate, mineral dust and sea salt. The M7 aerosol dynamics includes nucleation, coagulation, and sulfuric acid condensation processes. Coagulation and condensation lead to formation of mixed particles from the insoluble ones. Different aerosol types mentioned in above (as well as others, e.g. pollen particles) are provided as separate species in the model outputs along with lumped $PM_{10}$ and $PM_{2.5}$.

### b) Dry-deposition and Sedimentation

The dry deposition fluxes of gases and aerosols (for both number and mass concentrations) are calculated from the aerodynamic, quasi-laminar boundary layer as the product of the surface layer concentration and the dry deposition velocity (Stier et al., 2005). The fluxes are used as the lower boundary condition in the semi-implicit vertical diffusion TKE-CBR scheme (Cuxart et al., 2000). The calculation of the dry deposition velocities is performed by means of serial resistance approach. And the "big-leaf" method is used to calculate surface resistance (Ganzeveld and Lelieveld, 1995; Ganzeveld et al., 1998) per each grid-cell for the snow/ice, water, bare soil, low-vegetation and forest surface types. The $SO_2$ soil resistance is a function of soil pH, relative humidity, surface temperature, and the canopy resistance, while surface resistances for other gases are prescribed. The canopy resistance is computed from stomatal resistance and monthly mean Leaf Area Index (LAI) values from the Enviro-HIRLAM Interaction-Soil-Biosphere-Atmosphere scheme (Noilhan and Planton, 1989).

The sedimentation of the aerosol particles is calculated throughout the atmospheric column. The calculation of the sedimentation velocity is based on the Stokes velocity with the Cunningham slip-flow correction factor accounting for non-continuum effects (Seinfeld and Pandis, 2006). In order to satisfy the Courant-Friedrich-Lewy stability criterion, the sedimentation velocity is limited by ratio of the model layer thickness and the time-step.

### c) Wet-deposition

There are several options for the wet deposition in the model. The first version used the aerosol size dependent parameterisation of Baklanov and Sørensen (2001). In the latest version fixed size- and composition-dependent scavenging parameters are also applied for wet deposition calculation and are different for stratiform and convective clouds (Stier et al., 2005). They were derived from measurements of interstitial and in-cloud aerosol contents. These scavenging coefficients depend on the aerosol modes, total cloud water and fraction (liquid- and ice), and the conversion rates of cloud liquid water and cloud ice to precipitation through auto-conversion, aggregation, and accretion processes. The precipitation re-evaporation before it reaches the ground is also included. The STRACO cloud scheme (Sass, 2002) provides water- and ice- precipitation fluxes, normalized by the precipitation rates, to wet-deposition scheme, which uses prescribed size-dependent collection efficiencies for rain and snow (Seinfeld and Pandis, 1998).

### 2.5. Emission modules and pre-processor

The model includes anthropogenic, biomass burning (wildfires) and natural emission fluxes of both gases and aerosols. These emissions are processed in different ways; because some of them are pure datasets derived from ground-based and satellite observations. The others are interactively developing during the model integration and depend on the meteorological conditions at current time-step and land-use, -cover or water surfaces types. The anthropogenic emission inventory developed by TNO (Kuenen et al., 2014) and linked to the model is a dataset of yearly-accumulated fluxes of gases, such as CO, $CH_4$, $NO_x$, $SO_2$, $NH_3$, Non-Methane Volatile Organic Compounds (NMVOC), and particulate matter (PM) in two size bins – 2.5 μm and 10 μm, which are attributed to 10 source-sectors, e.g., energy industries, residential combustion, industry, etc., denoted by SNAP (Selected Nomenclature for sources of Air Pollution) codes. The inventory has resolution of $0.06^o$ x $0.12^o$ and covers the entire Europe, European part of Russia, North of Sahara and a part of Middle East. Total NMVOC emissions are split into 25 VOC compound groups by source-sectors by country (Kuenen et al., 2010). The $PM_{2.5}$, $PM_{10}$ emissions splitting into 6 aerosol species (BC, OC, Na, $SO_4$, Coarse Other Primary and Fine Other Primary particles) is applied following TNO recommendation (Kuenen et al., 2010). Because the dataset contains accumulated surface fluxes, one needs to redistribute them in order to reproduce diurnal, weekly and monthly emissions variability. The emissions can also occur at different heights, e.g., emissions from power plants are elevated and from traffic are at the surface; so, vertical redistribution is applied within first 8 model hybrid levels. Therefore, temporal and vertical profiles developed by TNO for different gaseous and aerosol species and SNAP codes are used in the emission pre-processor. The global biomass burning (wildfires) so-called the IS4FIRES (Sofiev et al., 2012) emission inventory developed by FMI has similar structure except a number and kinds of available gaseous and aerosol species as well as the resolution. The inventory data is total PM flux. The flux is split into $PM_{2.5}$ and coarse PM consisting of ash. The $PM_{2.5}$ primarily consists of Organic and Black Carbon (OC and BC) and a remainder of organic matter that is not carbon; for details see (Andreae and Merlet, 2001). The biomass burning emissions typically show a diurnal cycle variability, and therefore, corresponding coefficients are applied (Giglio, 2007). The wildfires emissions are also redistributed vertically having different proportions in lowest 200 m and the highest up to 1 km over the ground.

The natural emissions of gases and aerosols are fully interactive and calculated online. There is dimethyl sulfide (DMS; Nightingale et al., 2000) emission from oceans, which depends on the wind speed and seasonal variability of DMS solution in the water. Soluble sea-salt aerosol emissions (Zakey et al., 2008) are driven by wind speed and temperature and insoluble mineral dust aerosol emissions (Zakey et al., 2006) also depend on meteorology as well as hydrological parameters. Both sea-salt and dust aerosols are emitted in accumulation and coarse modes.

### 2.6. Aerosol feedback mechanisms

#### a) Direct and semi-direct effects

Enviro-HIRLAM contains parameterisations of the direct and semi-direct effects of aerosols. Direct and semi-direct effects are realised by modification of the Savijärvi radiation scheme (Savijärvi, 1990; Wyser et al. 1999) with implementation of a new fast analytical SW and LW aerosol transmittances, reflectances and absorptances. The 2-stream approximation equations for anisotropic non-conservative scattering described by Thomas and Stamnes (2002) are used for these calculations. The GADS/OPAC aerosols of Köpke et al. (1997) are used as input to the routine. The species include BC (soot), minerals (nucleus, accumulation, coarse and transported modes), sulphuric acid, sea salt (accumulation and coarse modes), "water soluble", and "water insoluble" aerosols. In addition to the more standard nucleation, accumulation and coarse aerosol size modes we consider, according to Köpke et al. (1997), the transported size mode to describe aerosols that have been transported over a long distance, for instance Saharan aerosols that have been blown to the Atlantic ocean. In order to make the calculations fast, optical properties that are spectrally averaged over the entire SW and LW spectra are used. The spectra used are shown in Figure 4. The short wave spectrum is a clear sky spectrum from 2 km height in a standard atmosphere (Anderson et al. 1986) calculated with the DISORT algorithm (Stamnes et al. 1988) run in the LibRadtran framework (Mayer and Kylling 2005). The long wave spectrum is calculated similarly and is based on the overall atmospheric LW transmittance of a standard atmosphere.

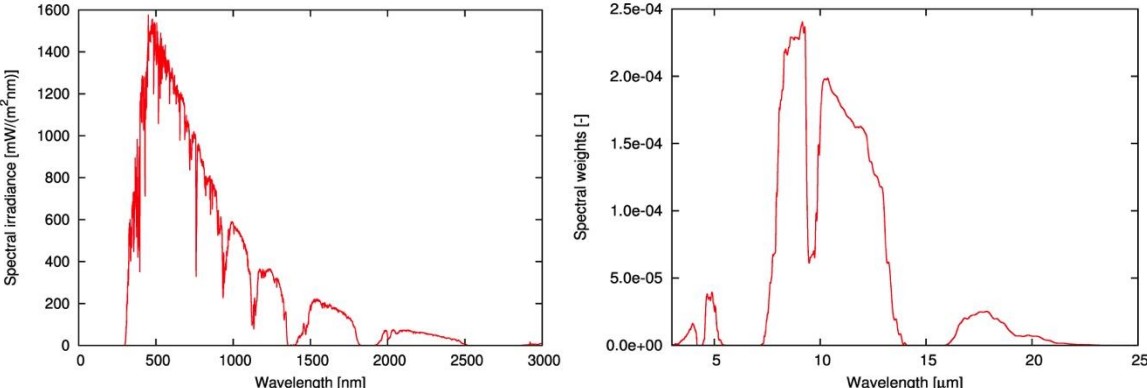

**Figure 4**: Left: The typical SW spectrum used for calculating average SW aerosol optical properties. Right: the spectral weights used for calculating average LW aerosol optical properties.

b) First and second indirect effects

For cloud-aerosol interactions a modified version of the Soft TRAnsition COndensation (STRACO) cloud scheme (Sass, 2002) is used in Enviro-HIRLAM. This scheme developed for operational NWP has recently been upgraded using new efficient methods to account for aerosol effects on cloud formation and microphysics. The scheme is able to account for convective transports of new variables. The prognostic aerosol fields are coupled directly to the cloud physical and microphysical properties. Liquid cloud droplet number is calculated based on aerosol size, number and solubility and the STRACO subgrid super saturation field is used as basis for the droplet nucleation calculation. This ensures consistency with the cloud water mass.

The modelled liquid droplet number evolves in time according to the following processes: droplet nucleation, self-collection, sedimentation and evaporation. In order to close the tendency calculations the liquid cloud droplet distribution is assumed to follow a gamma distribution where the shape parameter is calculated online using Geoffroy et al. (2010). Several schemes have been implemented for nucleation comprising Twomey (1959), Cohard et al. (1998), Cohard et al. (2000) and Abdul-Razzak et al. (1998), Abdul-Razzak and Ghan (2000). Self-collection is the process whereby droplets collide and stick together, but do not become rain-drops. The parameterization of self-collection processes follow Seifert and Beheng (2006). Sedimentation is calculated to be consistent with the mass of rain water in a given model time step under the basic assumption that the largest droplets are removed first from the cloud. Similarly, evaporation of a droplet below activation radius is calculated to be consistent with the total evaporated cloud water under the assumption that the smallest droplets evaporate first.
Cloud droplet effective radius controls the liquid phase absorptivity and transmissivity and is calculated from liquid water mass and droplet number and is here also dependent on the shape of the droplet distribution which evolves in time. Autoconversion follows Rasch and Kristjansson (1998), and is directly dependent on the calculated droplet number.
Abdul-Razzak and Ghan (2000) parameterization for aerosol activation has been extensively tested in many online-coupled weather and climate models. However, the STRACO cloud microphysics scheme with parameterizations of aerosol activation, cloud droplets nucleation, sedimentation, evaporation, self-collection, has been evaluated only with 1D column HIRLAM, so it needs to be further thoroughly evaluated.

### 2.7. Urban parameterisations and models urbanisation

The representation of urban areas in Enviro-HIRLAM contains the following aspects and processes (Baklanov et al., 2005):
(i) model down-scaling, including increasing vertical and horizontal resolution and nesting techniques; (ii) modified high-resolution urban land-use classifications, parameterizations and algorithms for roughness parameters in urban areas based on the morphologic method; (iii) specific parameterization of the urban fluxes in the meso-scale model; (iv) modelling/ parameterization of meteorological fields in the urban sublayer; (v) calculation of the urban mixing height based on prognostic approaches.
The nesting technics and downscaling methods are actively and successfully used for urban areas to reach the necessary resolution for resolving or parameterisation of urban features and effects. The details of this approach with the Enviro-HIRLAM model were described e.g. in Baklanov and Nuterman (2009). With respect to metropolitan areas, the downscaling for finer resolution allows to reproduce smaller scale meteorological patterns, and then these patterns are further modified through running urban parameterization modules only for grid cells where the cities are presented.
The urban parameterizations in the model contain three different approaches which may be combined. The first - simplest implementation contains modifications of the surface roughness, the anthropogenic heat flux, the storage heat flux and the albedo over urban areas. These are identified in the model using urban fractions extracted from the land-use database (CORINE) employed at DMI (Mahura et al., 2005b, 2006a, 2007a; Baklanov et al., 2005, 2008). The first module is the computationally cheapest way of "urbanising" the model and it can be used for operational NWP as well as for regional climate modelling. The second – Building Effect Parameterization (BEP) (Martilli et al., 2002) – module gives a possibility to consider the energy budget components and fluxes inside the urban canopy although it is a relatively more expensive (5–

410 10% computational time increase) (Mahura et al., 2008bc; 2010b; Figure 5). However, this approach is sensitive to the
411 vertical resolution of NWP models and is not very effective if the first model level is higher than 30 m. Therefore, the
412 increasing of the vertical resolution of current NWP models is required. The third – Soil Model for SubMeso Urbanized
(SM2-U) version (Dupont and Mestayer, 2006; Dupont et al., 2006) – module is considerably more expensive
computationally than the first two modules (Mahura et al., 2005a; Baklanov et al., 2008b). However, the third one provides
the possibility to accurately study the urban soil and canopy energy exchange including the water budget. Therefore, the BEP
scheme is considered as the baseline option and third SM2-U module is recommended only for use in advanced urban-scale
NWP and meso-meteorological research models. The details of implementations of different urban modules, own
developments and comparisons of different approaches and modules were published in previous papers (Mahura et al.,
2005ab; 2006a; 2008abc; 2010b; Baklanov et al., 2005, 2008b). The main approach includes an integration of the urban
modules into the ISBA (Interaction Soil- Biosphere- Atmosphere) land surface scheme of the NWP / HIRLAM model. The
urban modules are activated only on those grid cells of the model domain where the urban fraction is presented.

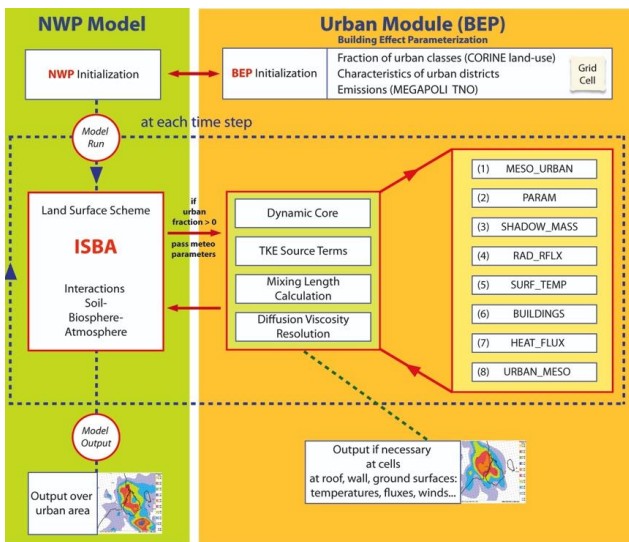

**Figure 5:** General scheme of the Building Effect Parameterisation (BEP) module for the Enviro-HIRLAM model
urbanization with a structure of the subroutine conception (adapted from Mahura et al., 2010b).
The urban boundary layer is very inhomogeneous and plays an important role in forming urban meteorological fields and
especially in dispersion of atmospheric pollutants, Therefore, for calculation of the urban mixing height, additionally to the
common diagnostic approaches, prognostic equations were used according to Zilitinkevich et al. (2002) and Zilitinkevich and
Baklanov (2002).
### 2.8. Transport schemes
Until 2012 there were basically two options for transport schemes in Enviro-HIRLAM (Chenevez et al., 2004): a) the
traditional non-conserving but highly efficient semi-Lagrangian (SL) scheme (Robert, 1981) in HIRLAM, b) the much less
efficient flux based and positive definite finite volume scheme by Bott (1989) with updates by Easter (1993). In 2012 the
default transport scheme was updated to a new monotonic version of the locally mass conserving semi-Lagrangian (LMCSL)
scheme (Kaas 2008, Sørensen et al. 2013). This scheme, used in the present version of Enviro-HIRLAM, to be described
briefly below is almost as efficient as the traditional SL scheme but now with the attractive properties of inherent mass
conservation, plus being monotonic and positive definite.
In HIRLAM and former versions of Enviro-HIRLAM a traditional SL scheme is used for advecting the specific
concentration of water constituents or the mixing ratio $q_i$ of any tracer $i$. Considering mixing ratio this means that when
ignoring any sources/sinks and turbulent mixing the prognostic transport equation to be solved is simply

$$\frac{dq_i}{dt} = 0 \qquad (1)$$

The traditional SL numerical integration of Eq (1) reads

$$(q_i)_k^{n+1} = (q_i)_{*k}^n \qquad (2)$$

where subscript $k$ is the grid point/cell index and superscripts $n$ and $n+1$ represent two consecutive time steps, respectively.
The subscript $*k$ indicates the tricubic interpolation to the location of the departure point of the upstream trajectory, which
arrives in grid point $k$ at time level $n+1$. The tricubic interpolation in (2) can also be represented as a sum of interpolation
weights involving 64 grid points surrounding the departure point. Formally this can be expressed

$$(q_i)_k^{n+1} = \sum_{l=1}^{K} w_{k,l} (q_i)_l^n \qquad (3)$$

where K is the total number of grid points in the entire integration domain. Note that for each $k$ only 64 $w_{k,l}$ weights are
different from zero. When converting mixing ratio into volume density, i.e., $(r_i)_k^{n+1} = (r_d)_k^{n+1}(q_i)_k^{n+1}$, and subsequently
summing over the integration area the traditional SL scheme is not mass conserving. Therefore in LMCSL (Kaas, 2008) a
different approach is followed, namely, as in most other mass conserving transport schemes, to solve the complete continuity
equation

$$\frac{\partial r_i}{\partial t} = -\nabla \cdot (r_i \boldsymbol{u}) \quad \text{or} \quad \frac{d r_i}{dt} = -r_i \nabla \cdot \boldsymbol{u} \tag{4}$$

still omitting sources/sinks and turbulent mixing an then evaluating the mixing ratio from $(q_i)_k^{n+1} = (r_i)_k^{n+1} / (r_d)_k^{n+1}$. In
LMSCL (4) is solved in a rather unusual way by modifying the interpolation weights in (3) in such a way that the sum of
mass given off at time step $n$ by a Eulerian grid cell $l$ to all departure points that it influences is exactly equal to its own mass.
In other words LMCSL is based on simple partition of unity. The modified weights become:

$$\hat{w}_{k,l} = \frac{V_l}{V_k} \frac{w_{k,l}}{\overset{K}{\underset{m=1}{\mathring{a}}} w_{m,l}} \tag{5}$$

where $V_k$ is the volume of Eulerian grid cell $k$. Using the modified weights the basic LMCSL forecast reads:

$$(\boldsymbol{\rho}_i)_k^{n+1} = \sum_{l=1}^{K} \hat{w}_{k,l} (\boldsymbol{\rho}_i)_l^n \tag{6}$$

As the traditional SL scheme the LMCSL is not inherently monotonic or positive definite. Therefore an a posteriori iterative
locally mass-conserving (ILMC) filter was developed, Sørensen et al. (2013). This filter ensures that the mixing ratio of the
forecast will never be larger/smaller than the largest/smallest mixing ratio of the eight grid cells surrounding the upstream
trajectory departure point at time level $n$. The ILMC filter designed to be as local as possible since non-local filters will
generate non-physical chemical reactions. This is ensured by an iterative approach where the mass discrepancy is re-
distributed among the neighbouring cells in the first iteration, and increasing the distribution radius, in case there is remaining
mass discrepancy, for the next iteration(s). In general one or two iteration(s) are sufficient.

The LMCSL transport scheme in combination with the ILMC produces accurate monotonic and positive definite forecasts for
water vapour, liquid/ice water and chemical constituents. As an example the simulated $PM_{2.5}$ concentration on July 17 in
2010 with horizontal resolution of approximately 16 km's is shown in Figure 6. It can be seen that the model is able to
reproduce, e.g., sharp transitions related to fronts over the North Atlantic. A more in depth analysis of the ability of ILMC to
reproduce sharp gradients can be found in Sørensen et al. (2013), in particular Figure 3 and the accompanying discussion in
that paper.

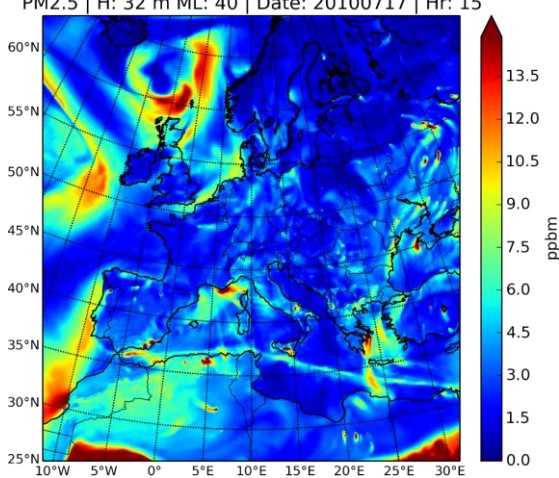

**Figure 6:** Example of the simulated $PM_{2.5}$ concentration over Europe on July 17 in 2010 with horizontal resolution of 16 km.

It should be noted that the dynamical core in Enviro-HIRLAM is identical to that of HIRLAM. Thus, the dry-air density for
dynamics is calculated using a traditional SL approximation to (Eq. 4), i.e. not the LMCSL. Therefore, the Enviro-HIRLAM
is not formally wind-mass consistent regarding tracer transport. However, the large scale precipitation fields in the traditional
HIRLAM and Enviro-HIRLAM are very similar (see, e.g., Figure 4 in Sørensen et al. (2013)), which suggests that wind-mass
inconsistency is of minor importance. In principle no monotonic transport schemes can be mass-wind consistent since the
monotonic limiters formally destroy the consistency (see discussions on the issue of mass-wind inconsistence in atmospheric
models in Jöckel et al. (2001)).
**3. Modelling system applications**

Possible applications of the online integrated Enviro-HIRLAM modelling system include the following: chemical weather forecasting, air quality and chemical composition longer-term assessment, weather forecast (e.g., in urban areas, severe weather events, etc.), pollen and bio-aerosols transport forecasting, climate change modelling, studies of climate change effects on atmospheric pollution on different scales, anthropogenic impacts on atmospheric processes, weather modifications, geo-engineering, contamination from volcano eruptions, sand and dust storms, nuclear explosion consequences, and other emergency preparedness modelling.

Several realised/tested types of applications of the Enviro-HIRLAM for meteorological, environmental and climate forecasting and assessment studies are highlighted in Figure 1 and will be demonstrated below.

### 3.1. Applications for Numerical Weather Prediction

Several Enviro-HIRLAM sensitivity and validation studies of aerosol feedbacks on meteorological processes were done previously (see e.g., Korsholm, 2009; Korsholm et al., 2010; Baklanov et al., 2011ab; Sokhi et al., 2016). For example, the effects of urban aerosols on the urban boundary layer height, can be comparable with the effects of the urban heat island ($\Delta$h is up to 100–200m for stable boundary layer) (Baklanov et al., 2008a). Further studies (Korsholm et al., 2010) of megacities effects on the meteorology/climate and atmospheric composition showed that aerosol feedbacks through the first and second indirect effect induce considerable changes in meteorological fields and large changes in chemical composition (see Section 3.4), in a case of convective clouds and little precipitation. The monthly averaged changes in surface temperature due to aerosol indirect effects of primary aerosol emissions in Western Europe were analysed and validated vs. measurement data. It was found that a monthly averaged signal (difference between runs with and without the indirect effects) in surface temperature can reach 0.5°C (Figure 2.2b in Korsholm et al., 2010). Korsholm (2009) studied the impact of aerosol indirect effects on surface temperatures and air pollutant concentrations for a 24 h simulation over a domain in northern France including Paris in a convective case with low precipitation. He found a marginally improved agreement with observed 2m temperatures and a marked redistribution of $NO_2$ in the domain, primarily as a result of the second indirect effect.

To perform analysis of atmospheric aerosol effects on clouds and precipitation, the year 2010 was selected for Enviro-HIRLAM simulations. That year, especially summer, was characterized by severe weather events such as floods, heat waves and droughts across Middle East, most of Europe and European Russia. The model was forced by boundary and initial conditions produced by ECMWF IFS (IFS-CY40r1) and MOZART (Horowitz et al., 2003) models for meteorology and atmospheric composition, respectively. The Enviro-HIRLAM modelling domain with horizontal resolution of 0.15° x 0.15° having 310 x 310 grid cells, and 40 vertical hybrid sigma levels extending to pressures less than 10 hPa, covers Europe, North of Sahara, and European Russia. The modelling domain was partitioned into 120 CPU cores and the model was run with time step of 300 seconds. The model includes emissions from anthropogenic sources developed by TNO and from wildfires produced by FMI as well as interactive DMS, sea-salt and dust emissions (for details see Sect. 2.5).

For aerosol-cloud interactions, these were estimated also for July 2010 by means of delta function, i.e., difference between outputs of models: Enviro-HIRLAM with aerosol-cloud interactions (ENV) and Reference-HIRLAM (REF). Fig. 7a shows deltas (ENV–REF) of total cloud cover over model domain, which is mainly increased (with local maxima up to 90%) except several inland areas, such as Finland, borders of Germany, Poland and Austria, where cloud cover decreased by almost 10 fold. The ENV runs revealed the increase of average cloud top height by approximately 2%. The delta function of cloud water content at average cloud base shows (Fig. 7b) its increase compared to REF and local maxima over North Atlantic, North Sea, Sweden, Switzerland, and Austria. These areas are occupied by precipitating clouds as seen in Fig. 8.

The absolute frequencies of stratiform and convective precipitation over computational domain are decreased compared to the REF model, while the amount of convective precipitation during heavy precipitation events is increased. Hence, the wet deposition of particles decreases in summer because it rather depends on precipitation frequency than on its amount. The REF model run tends to over-predict both frequency and amount of precipitation. But the inclusion of aerosol-cloud interactions can improve general model performance, i.e., the ENV run bias for precipitation with respect to its frequency and amount has been decreased compared to the REF model run (Fig. 9).

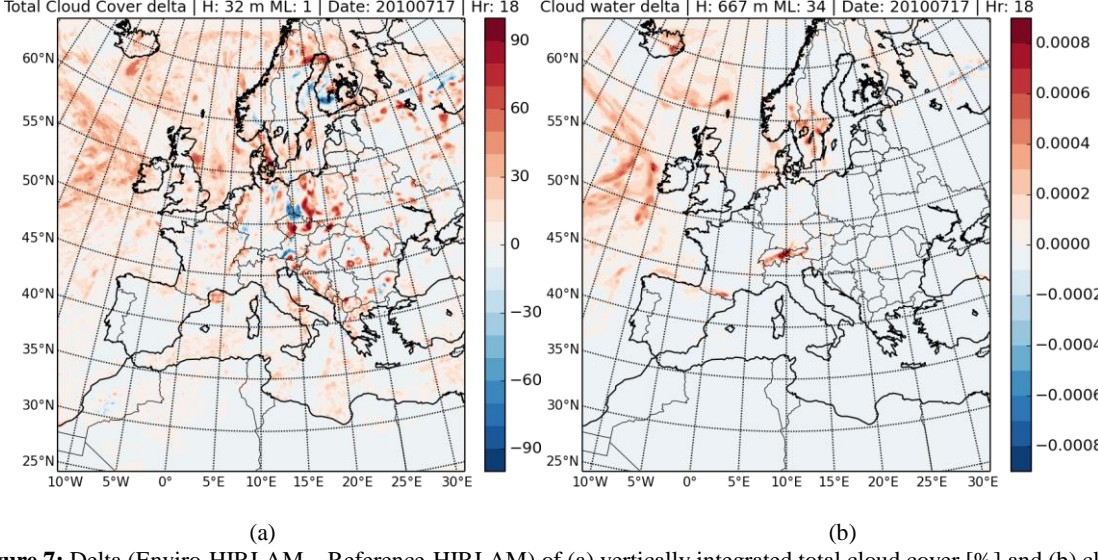

(a)                                                                              (b)

**Figure 7:** Delta (Enviro-HIRLAM – Reference-HIRLAM) of (a) vertically integrated total cloud cover [%] and (b) cloud water content [kg/kg] at average cloud base (667 m) on 17 Jul 2010, 18 UTC.

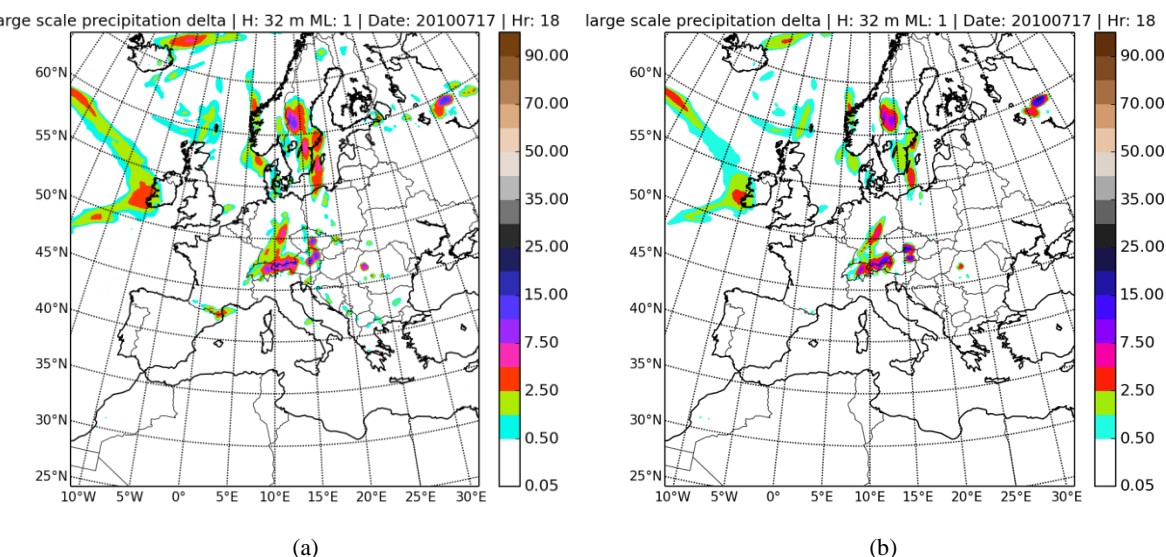

(a)                                                                              (b)

**Figure 8:** Accumulated (3 hour) precipitation patterns from Reference-HIRLAM (REF) and Enviro-HIRLAM with aerosol–cloud interactions (ENV) on 17 Jul 2010, 18 UTC: stratiform precipitation: (a) – REF, (b) – ENV.

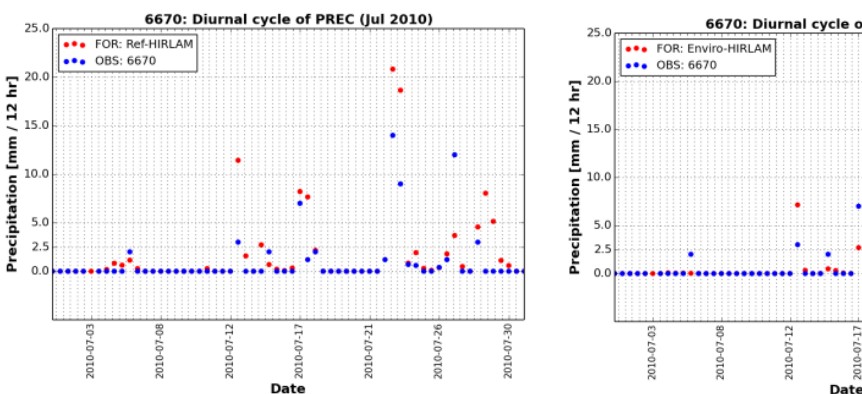

**Figure 9:** Precipitation amount (12 hours accumulated) of reference HIRLAM (left) and Enviro-HIRLAM with aerosol–cloud interactions (right) vs. surface synoptic observations at WMO station 6670 at Zurich, Switzerland (lat: 47.47; lon: 8.53) during Jul 2010.

Sensitivity studies on the model response to aerosol effects indicate strong "signals", but it doesn't guaranty improvements.
E.g., Korsholm (2009) considered evaluations only for some elements (e.g., the coupling interval) in the previous analysis
and made corresponding conclusions about the improvements. Other feedback mechanisms, especially for aerosol-cloud
interactions, were analysed mostly as sensitivity studies or evaluations for short-term episodes.
The model formulations have only been tested on a case basis and although strong signals have been found, this does not
imply improved meteorological performance of the model. In particular, testing over longer periods including all seasons was
not conducted that time. Furthermore, the interactions between aerosols and the cloud ice-phase are not in a state where
improvements would be expected. Therefore, it is necessary to mention that it is too early to make conclusions about the
improvement of precipitation forecasting by implementation of the indirect aerosol effects, because of large uncertainties in
parameterisation of the cloud-aerosol microphysics processes (especially for ice-nucleation) and due to adjustments of such
effects indirectly in NWP model parameters and constants (retuning of them after implementation of the aerosol feedbacks is
needed). More investigations, further improvements and evaluations are needed for aerosol indirect effects and aerosol-cloud
microphysics schemes in the model. Recently such evaluation studies are realised within the CarboNord project for monthly
and annual validation studies and will be published separately.

### 3.2. Urban meteorology and environment prediction and assessments

The analysis of urban boundary layer (UBL) for metropolitan areas of megacity Paris (more than 10 mil population) and
growing medium-size Bilbao (1 mil) placed over a semi-flat and coastal-complex terrains, respectively, was performed
employing the Enviro-HIRLAM model. In particular, the 1) evaluation of the model performance coupled with urban module
for different types of terrain and size of cities; and 2) estimation of urban heat island (UHI) development over selected urban
areas and surroundings were done.

The Enviro-HIRLAM simulations were performed for nested domains with horizontal resolutions of 15, 5 and 2.5 km and for
selected periods in July 2009. The meteorological boundary conditions were provided by the European Centre for Medium
Range Weather Forecast (ECMWF) every 3 hour. The model was employed in 2 modes. The 1st mode is *control (CTRL)* run.
The 2nd mode is *urban (URB)* run – e.g. coupled with the Building Effect Parameterization (BEP, Martilli et al., 2002)
module and anthropogenic heat fluxes (AHF) from the Large scale Urban Consumption of energY (LUCY) model (Allen et
al., 2010). Extracted AHFs were 60 and 40 W m$^{-2}$ for the Paris and Bilbao metropolitan areas, respectively. For the URB run
at the finest resolution, the Paris and Bilbao urban areas were represented by 220 and 16 urban cells, respectively (Figure 10;
adapted from González-Aparicio et al. 2010). In each grid-cell, BEP parameterizes the flux exchange between the urban
surface and the atmosphere depending on combination of different urban districts, e.g. residential, low and high buildings,
industrial and commercial.

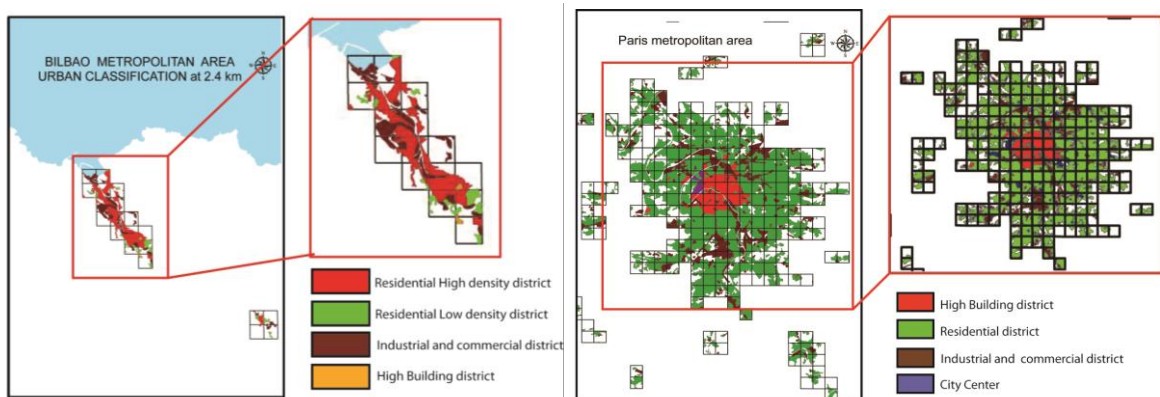

**Figure 10**: Urban district classification based on urban zoning data for the a) Bilbao and b) Paris metropolitan area, including
the residential area (ReD), low and high building districts (LBD and HBD, respectively) and industrial and commercial
districts (ICD). Spatial distribution of urban districts (HBD – high buildings, RD – residential, ICD – industrial commercial,
and CC – city center districts) for the Paris metropolitan area within the P01 modelling domain (partly adopted from
González-Aparicio et al. 2014).

The statistical analysis showed that the urban simulation had a reduced bias with respect to observations than the control
simulations. For Paris, on a monthly basis, the correlations for air temperature were higher for the URB compared to CTRL
run, and results improved up to 10% on a diurnal cycle (with a maximum of 0.83 at 08 UTC). The correlations were slightly
lower (down to 0.5) at early morning hours and slightly higher (up to 0.8) during afternoon and night-time. Moreover,
correlations at suburban and urban stations were similar to correlations at rural stations (see Figure 11a). Analysis for Bilbao
(González-Aparicio et al. 2013) showed similar performance of the model for both runs: with correlation for air temperature
about 0.85 and 0.88 for summer and winter, respectively. For the specific humidity it was 0.75 and 0.92. For the wind speed,
the highest value (0.8) is in summer, and during winter it decreased to 0.6 (0.4) near the coast (inland) stations.

The results of simulations for two selected cities showed that the model reproduced well the meso-scale processes at regional scale, inland winds over Paris and land-sea breeze interactions over Bilbao. For selected locations (e.g. coastal vs. inland sites), the bias between the observations and simulations was higher over Bilbao (maritime) than over Paris (continental) cities. Although hydrostaticity of the model over a complex terrain is a limitation, but sensitivity test over Bilbao showed that at 2.5 km optimal resolution it is possible at the same time to obtain satisfactory reproducibility of the large scale processes and to explore the urban effects at finer scales.

The UHI development was also for short-term periods (here, for Paris – 28 Jul 2009; for Bilbao – 15 Jul 2009) with calm and anticyclonic conditions. For Paris, three different locations were considered: urban (LHVP), suburban (SIRTA) and rural (CHARTRES) stations (Figure 11a). As seen, the UHI was fully developed at 04 UTC with air temperature anomaly of 2.2℃ (LHVP) and 0.6℃ (SIRTA). It started at mid-night and expanded covering area of about 2000 km$^2$. The heat island was retained until 11 UTC, but during the daytime (e.g. 11-17 UTCs) the effect disappeared due to contribution of incoming solar radiation. At CHARTRES this effect (0.2℃) was almost negligible. Both the wind speed and relative humidity were also affected by the urban area: at LHVP the wind speed reduced by maximum 3.5 m s$^{-1}$ at 06 UTC, and the relative humidity - down to 15% under developing UHI. At SIRTA the change in wind speed was down to 0.7 m s$^{-1}$ and at CHARTRES the changes in wind speed and relative humidity were almost negligible.

For Bilbao, model showed that for breezes from northern directions, the impact of urban area on local flow dynamics is inhibited; however, for breezes from southern directions - the urban effect had appeared. For example, on 15 Jul 2009, the UHI was developed during night-morning hours (e.g. 23-09 UTCs) with maximum up to 1℃, and heat island expanded covering area of about 130 km$^2$. In addition, González-Aparicio et al. (2013) showed that the UHI intensity is lower in winter compared with summer, underlying that dominating factor is the surface heating during daytime, which is higher in summer than in winter.

As medium size cities are under continuous development, future impacts of urbanization are expected to become more significant. Several different scenarios of urban development were tested for Bilbao (González-Aparicio et al., 2014). Enviro-HIRLAM model runs showed that under calm conditions during summer and winter, the UHI could reach up to 2.2℃ covering area of about 400 km$^2$ when city is doubled in size or doubled in AHF. When city is tripled in size, the UHI could reach up to 3℃ with urban island expansion up to 550 km$^2$ (Figure 11c). Analysis of UHI for Bilbao (e.g. triple size city scenario) vs. current UHI over Paris showed similar intensity of up to 3℃, and UHI boundaries are different, e.g. for Paris it was 4 times larger. Such differences can be explained by different cities' sizes, morphologies and characteristic AHFs.

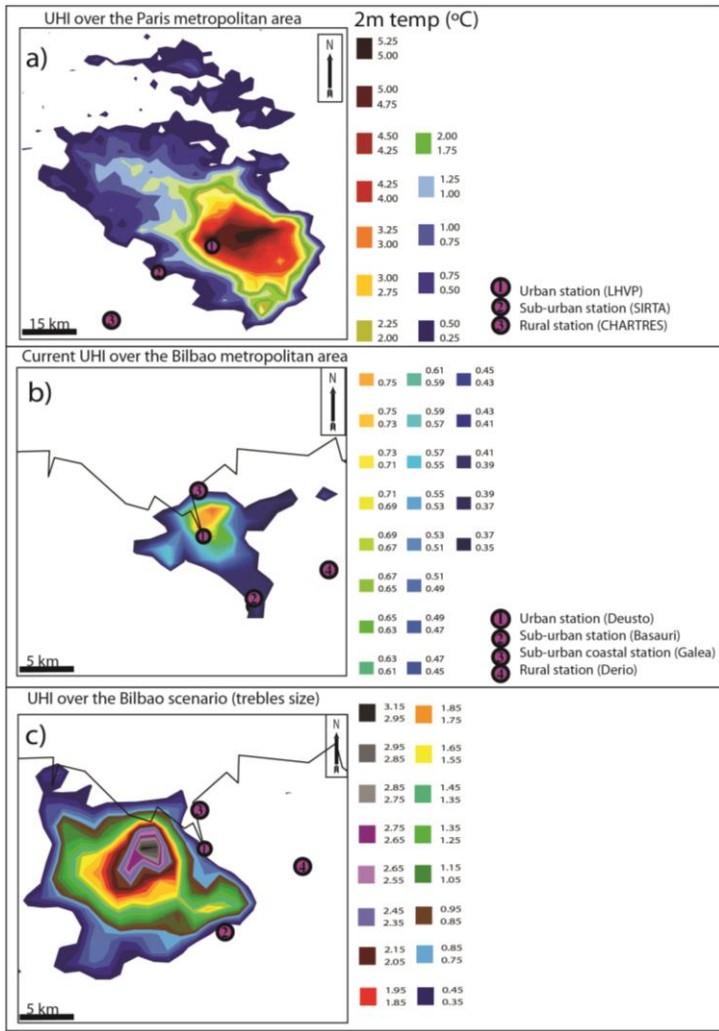

**Figure 11**: Difference plots for the air temperature at 2m between outputs of the URB (urbanized -BEP + AHF-) and CTRL (non-urbanized) Enviro-HIRLAM model under calm conditions during summer 2009 for the a) Paris metropolitan area and for the Bilbao metropolitan area b) in its current size of the city and c) under a scenario tripling the size of the city.

### 3.3. Pollen forecasting

Among air-pollinated allergens, birch pollen is one of the most important for the population group suffering allergic diseases. The number of allergic patients sensitive to birch pollen is assessed as 20% of European population (WHO, 2003; Linneberg, 2011) and this number is constantly increasing. In particular, in Denmark the number of allergic patients has increased twice over the past few decades (Linneberg, 2011). These facts demonstrate the importance of operational birch pollen forecasting for the European population especially during the spring season. Currently, birch pollen is presented as biological air pollutant in different NWP and ACT models such as SILAM (Finland), COSMO-ART (Germany, Switzerland, Austria), CHIMERE (France), Enviro-HIRLAM, DEHM (Denmark) and others. The pollen emissions are strongly dependent on meteorology, so it is advantageous to simulate and forecast pollen pollution episodes by online-coupled meteorology-air pollution models since all necessary meteorological fields are available at each model time step.

Original developments of the dynamical Enviro-HIRLAM based operational modelling system for the birch pollen forecasting in Denmark (called Env-POLL) were started in 2006 (Rasmussen et al., 2006; Mahura et al., 2006b) including previously developed statistical methods (Rasmussen, 2002), modelling of elevated concentrations episodes, analysis of spatio-temporal and diurnal cycle variabilities, contribution of remote source regions into pollen levels, improvements in emissions and parameterizations, etc. (Mahura et al., 2007b, 2009, 2010a). The most recent developments are shown in Kurganskiy et al. (2015) with revised general scheme of input and output of the Enviro-HIRLAM birch pollen forecasting system presented in Figure 12. The input includes the meteorological initial/ boundary conditions (IC/BC) obtained from the IFS model system, birch forest fraction map, phenological data, i.e. temperature sum thresholds for start of flowering (Sofiev et al., 2013), accumulated total number of birch pollen particles emitted from a unit area during the pollinating season.

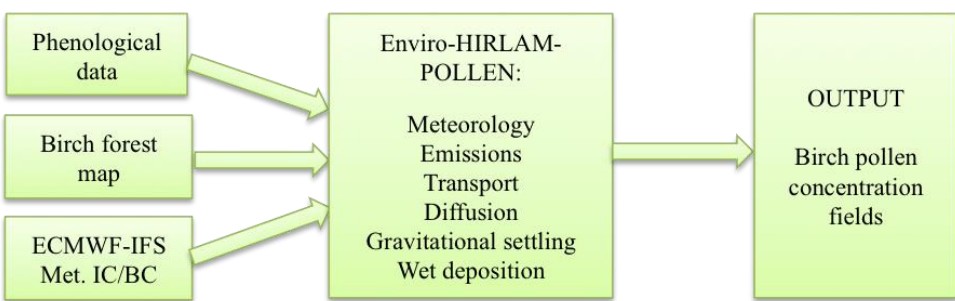

**Figure 12**: General scheme of Enviro-HIRLAM birch pollen forecasting.

The forecasting of birch pollen concentrations requires information/data on the spatial birch tree distributions, characteristics of pollen release, its atmospheric transport and dispersion, its deposition due to gravitational settling and wet deposition, i.e. scavenging by precipitation. Birch pollen emissions are fully dependent on temporal and spatial variability of meteorological conditions. The emission module (Sofiev et al., 2013) includes the following parameters affecting the pollen release: 2-meter air temperature and relative humidity, 10-meter wind speed, and accumulated precipitation. The atmospheric transport is handled in the same way as for aerosols (see section 2.8). Dry deposition of birch pollen particles in the atmosphere is represented by gravitational settling (Seinfeld and Pandis, 2006) whereas dry deposition due to interactions of particles with the surface can be neglected according to Sofiev et al., (2006). The wet deposition scheme distinguishes between in-cloud (Stier et al., 2005) and below-cloud scavenging (Baklanov and Sørensen, 2001). The output in terms of birch pollen forecasting, and for analysis, contains 2D fields of the birch pollen concentration at the lowest vertical model level. The modelling domain has 15 km horizontal resolution with 154 and 148 grid points along longitude and latitude, correspondingly. The domain covers the main European part and is centered around Denmark.

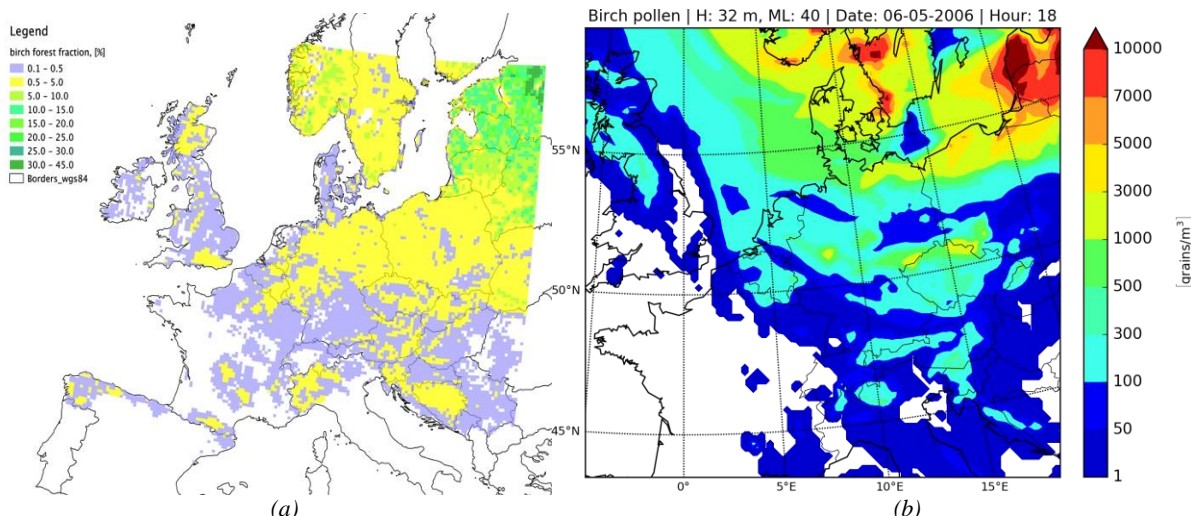

*(a)*                                    *(b)*

**Figure 13**: (a): Birch forest fraction map; (b): Example of the simulated birch pollen concentration in the modelling domain on the 6[th] of May, 2006 at 18 UTC.

Birch forest habitat map has been derived by GIS (Geographic Information System) analysis (http://www.spatialanalysisonline.com) for the selected modelling domain. The map (Fig. 13a) shows birch forest fraction in each model grid cell. Three GIS based databases were used in the derivation procedure: 1) Global Land Cover Characterisation (GLCC, http://landcover.usgs.gov/glcc/), 2) European Forest Institute (EFI, Päivinen et al., (2001)) and 3) Tree Species Inventory (TSI, Skjøth et al., (2008)). Both GLCC and EFI have 1 km horizontal resolution, whereas TSI has 50 km resolution.

As examples for the birch pollen season 2006 the model results were compared with observations for two Danish sites: Copenhagen and Viborg (Fig. 14). This year was dominated by a relatively cold spring over large areas of Europe followed by rapid warming and little/no rain. It caused short but intensive birch pollen season with long range transport episodes before the local flowering start and thereby emissions. The evaluation for both modelled and observed birch pollen concentrations showed extremely high values (daily averages about and even more than 1000 grains/m$^3$) during 5-10 May 2006 episode for Copenhagen and 5-8 May 2006 episode for Viborg. The extremely high birch pollen concentrations over Denmark are also visible in Fig. 13b.

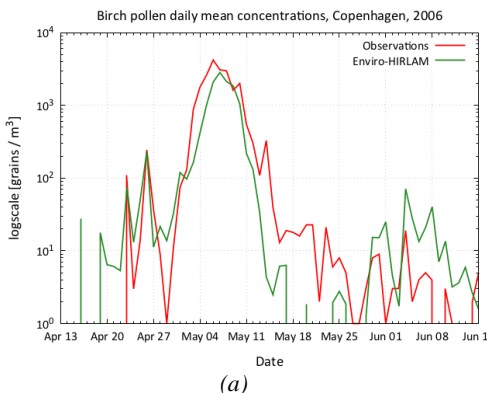
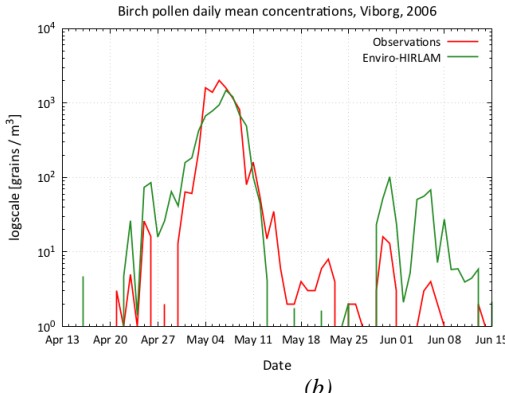

|     |     |
| --- | --- |
| *(a)* | *(b)* |

**Figure 14:** Birch pollen concentrations observed (red) vs. modelled (green) at Danish sites: Copenhagen (a) and Viborg (b).

According to Sofiev et al., (2011) and Siljamo et al. (2013) the following criteria can be used for assessment of birch pollen concentration forecasting: model accuracy (MA), hit rate (HR), false alarm ratio (FAR), probability of false detection (POFD) and odds ratio (OR). All of the criteria are calculated using four parameters obtained by assessment of the number of low and high modelled vs. observed birch pollen concentrations (C) relatively to a threshold value $N_{th} = 50$ grains/m$^3$ (i.e. C $\geq N_{th}$ for high and C $< N_{th}$ for low-concentration days). The threshold has been chosen since most of the pollen allergy sensitive population might start suffering from allergic reactions when daily mean birch pollen concentration, C $\geq N_{th}$ in the air (Jantunen et al., 2012).

The results of statistical analysis showed high MA for both Danish stations (0.95 for Copenhagen and 0.84 for Viborg, 0.9 in average). Prediction of elevated/top concentrations (HR values) by the model was assessed as 0.93 for Copenhagen and 0.58 for Viborg. The FAR values indicated that the probability to get an incorrect top model concentration was 0.07 and 0.42 for Copenhagen and Viborg, respectively. The POFD criterion showed low probability to get high modelled concentrations for observed low-concentration days (0.02 for Copenhagen and 0.18 for Viborg). Finally, the OR indicated that the likelihood for getting "high" day concentration instead of a "low" (if the model prediction is "high") were 42 and 3.26 times higher for Copenhagen and Viborg, respectively. In other words, the OR values show the ratio between HR and POFD. As it is seen from the OR values provided above, a fraction of the correct forecasts is prevailing for both Danish stations in this study.

It was found that comparing with observations, the modelled results reflected the general shape of changes in pollen concentration during the episode studied for both Danish stations: Copenhagen and Viborg. As it is also seen in Fig. 14 the model reproduces the magnitude of birch pollen concentrations for the peak period of the season in comparison with observations. However, some overestimation of the modelled concentration is visible for both stations at the end of the season. It can be explained by contribution due to long-range atmospheric transport of pollen from other remote regions, presumably from those located more northerly than Denmark and where the pollen season starts and ends later relatively to the Danish sites.

### 3.4. Chemical Weather Forecasting and air pollution applications

Validation and sensitivity tests (on examples of case studies and short-time episodes) of the online vs. off-line integrated versions of Enviro-HIRLAM (Korsholm et al., 2008) showed that the online coupling improved the results. Different parts of the model were evaluated vs. the ETEX-1 experiment, Chernobyl accident and Paris MEGAPOLI campaigns (summer 2009) datasets and showed that the model had performed reasonably well (Korsholm, 2009; Korsholm et al., 2009; 2010; Sokhi et al., 2017).

Online vs. off-line coupled simulations for the ETEX-1 release showed that the off-line coupling interval increase leads to considerable error and a false peak (not found in the observations), which almost disappears in the online version that resolves meso-scale influences during atmospheric transport and plume development (Korsholm et al., 2009). Further studies (Korsholm et al., 2010) of urban aerosol effects on the atmospheric composition showed that aerosol feedbacks through the first and second indirect effect induce large changes in chemical composition, in particular nitrogen dioxide, in a case of convective clouds and little precipitation. For the Paris campaign, on diurnal cycle variability the ozone concentration patterns showed dependencies on meteorological parameters, and especially seen at urban scale runs (Mahura et al., 2010b).

To perform further analysis of online coupling and feedback effects on atmospheric pollution forecasting, the year 2010 was selected (for details see Sect. 2.5). Nuterman et al. (2013) evaluated the Enviro-HIRLAM model for July 2010 vs. ground-based observations of PM$_{2.5}$ from EU AirBase air-quality network (Guerreiro et al., 2014), with a number of stations located in Denmark, Sweden, Germany and Spain (see Fig. 15a). The model runs were performed for the entire July 2010 with 7 days spin-up in June. Fig. 15b shows correlation coefficients on a diurnal cycle for PM$_{2.5}$ concentrations at selected measurement sites. In general it shows a fairly good positive correlations (more than +0.3), except for several Spanish stations (such as ES1938A at daytime, and ES1974A - at nighttime).

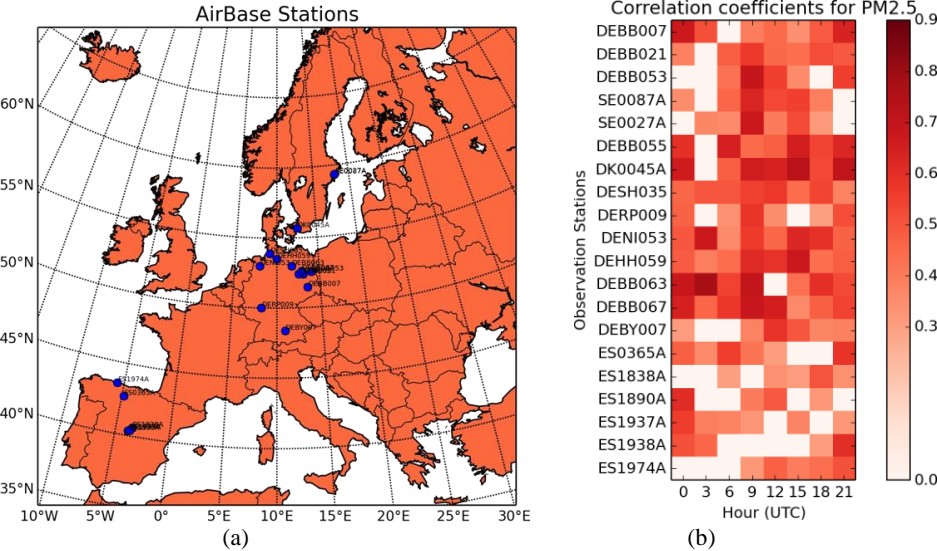

(a)                                                              (b)

**Figure 15:** (a) Map of selected AirBase air-quality monitoring stations (http://acm.eionet.europa.eu/databases/airbase/) across Europe; (b) $PM_{2.5}$ correlation coefficient on diurnal cycle for selected AirBase observation stations.

On the monthly based evaluation the model predicts well $PM_{2.5}$ day-to-day variability, but always has negative bias (Fig. 16). This under-prediction is due to several reasons: i) aerosol microphysics without secondary organic aerosols; ii) lack of partitioning of ammonium nitrate; iii) rough model resolution, which still cannot capture small-scale effects like complex orography and urbanized regions (in particular, due to lack of fine-resolution emissions from anthropogenic sources, like urban traffic). For instance, the model shows negative bias of $PM_{2.5}$ during daytime at Danish urban station (Fig. 16a). It is apparently due to rough model resolution in the considered runs. It was also found that $PM_{2.5}$ values are very influenced by changes in atmospheric stability conditions, which difficult to predict accurately in many NWP models. This can be observed from correlation coefficient decrease at stations during night-time (at 03 UTC) or from underestimation of elevated concentrations. In spite of these issues, the model can well reproduce diurnal cycle of aerosols at different sites, e.g. urban (Fig. 16a), coastal and rural (Fig. 16b), and shows good overall performance.

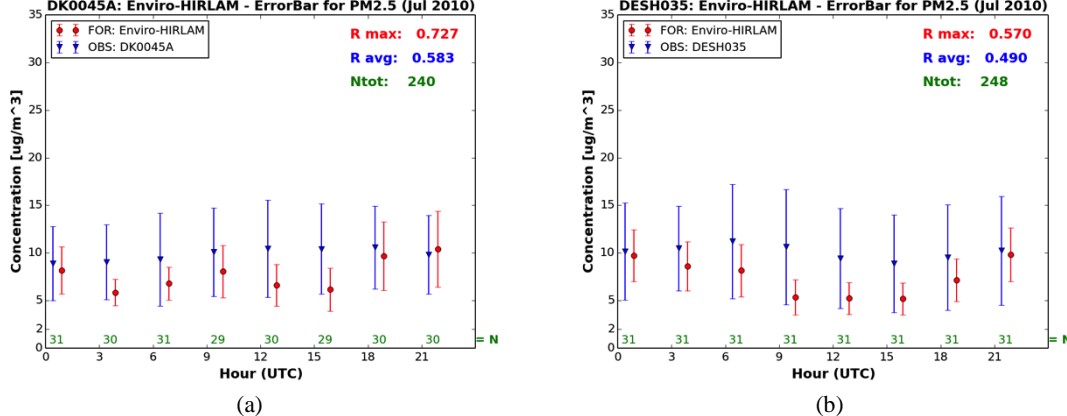

(a)                                                              (b)

**Figure 16:** Error-bar concentrations [ug/m$^3$] on diurnal cycle for AirBase observations vs. Enviro-HIRLAM modelling results; (a) Danish urban station and (b) German rural station; Right top corner indicates maximum and average correlation coefficients for the station as well as total number of analysed observation samples; Green numbers along X axis indicate number of observation samples per time slice.

Further on-going developments of the Enviro-HIRLAM modelling system for atmospheric composition applications are realised within the FP7 MarcoPolo and NordForsk CarboNord projects. The Enviro-HIRLAM downscaling from regional-to-urban scale modelling is realised in MarcoPolo for the East China region and largest metropolitan agglomerations in China (Mahura et al., 2016) with a focus on providing services on meteorology and atmospheric composition (with focus on aerosols). The Northern Hemispheric low resolution modelling in a long-term mode is realized in CarboNord with focus on evaluation of black carbon as well as higher resolution modelling over European domain in a short-term mode with focus on feedbacks mechanisms evaluation (Nuterman et al., 2015; Kurganskiy et al., 2016).

## 4. Further discussions

Several types of the above described and previously published applications of the Enviro-HIRLAM for meteorological, environmental and climate forecasting and assessment studies were tested and demonstrated. Different applications of Enviro-HIRLAM (with downscaling from hemispheric - regional – subregional – urban scales) were realised for different geographical regions and countries including European countries such as Denmark, Lithuania, France, Spain, Ukraine, Russia, The Netherlands, Turkey and well as for other – Kazakhstan, China and Arctic regions.

It is clear that the seamless/ online integrated modelling approach realised in Enviro-HIRLAM is a perspective and state-of-the-art way for future single-atmosphere modelling systems, providing advantages for all three communities: meteorological modelling including NWP, AQ modelling including CWF, and climate modelling.

However, there is no necessarily one configuration of the integrated online modelling approach/ system suitable for all communities, and that should be further investigated with practical needs for areas applications, approaches to coupling and computing resources usage. In particular, based on previous studies and above shown examples the following could be recommended for the considered applications:

- For AQ: online coupling improves air quality forecasts, and especially with full chemistry and aerosol feedbacks effects included.
- For NWP: gas chemistry is not critical and can be simplified (or omitted), but aerosol feedbacks are important for radiation and precipitation, and especially, for heavy polluted episodes and in urban areas.
- For pollen forecast: online coupling improves pollen emission parameterization and correspondingly modelling of concentration and deposition; however, the feedbacks are not so important; the chemistry is not considered yet, but interaction with allergens would be interesting to study in future (not done yet).
- For climate studies: it is suitable only for understanding the feedback mechanisms, but too expensive computationally for climate time-scale runs (the model had been used usually for one year period runs); chemistry is important, there is a need to be optimised and simplified.

It should also be mentioned that the considered evaluations of Enviro-HIRLAM were done only for some elements (e.g., the coupling interval) in the previous analysis and main conclusions about the improvements were provided just for these. Other feedback mechanisms, and especially for aerosol-cloud interactions, were analysed mostly as sensitivity studies or evaluated for short-term episodes. In particular, the STRACO cloud scheme contains fairly simplified cloud microphysics (heavily parameterized). Hence, tuning is essential for the overall performance of the model, when it comes to precipitation and cloud physical properties.

## 5. Conclusions

In this manuscript we have provided a comprehensive description of the Environment – HIgh Resolution Limited Area Model (Enviro-HIRLAM), which is developed as a fully online coupled/ integrated numerical weather prediction and atmospheric chemical transport modelling system for research and forecasting of joint meteorological, chemical and biological weather.

Possible applications of the modelling system can include: chemical weather forecasting, air quality and chemical composition for short- and long-term impact assessments on population and environment, multi-scale weather forecasting (e.g., on regional and subregional scales, in urban areas, severe weather events, etc.), pollen and road weather conditions forecasting, climate change forcing modelling, studies of climate change effects on atmospheric pollution on different scales, weather modification and geoengineering methods, forest fires and volcano eruptions, dust storms, nuclear explosion consequences, other emergency preparedness modelling.

Comprehensive online modelling systems, like Enviro-HIRLAM, built originally for research purposes and including all important mechanisms of interactions, will help to understand the importance of different physical-chemical processes and interactions and to create specific model configurations that are tailored for their respective purposes.

Multiple episode studies with the Enviro-HIRLAM model demonstrated the importance of including the meteorology and chemistry (especially aerosols) interactions in online-coupled models. However, there is no one unique integrated online modelling system configuration, which is the best suitable for all communities.

Highlighting a number of previous investigations we show that Enviro-HIRLAM has already been used for a host of different applications ranging from pollen forecasting to numerical weather prediction.

It should be stressed that there are still main gaps remaining in understanding of several processes such as: (i) aerosol-cloud interactions (still poorly represented); (ii) data assimilation in online models (still to be developed to avoid over-specification and opposite cancelling effects); and (iii) model evaluation for online models needs more (process) data and long-term measurements – and a test-bed.

**Code and/or data availability**

The Enviro-HIRLAM modelling system is a community model. The source code is available for non-commercial use (i.e. research, development, and science education) upon agreement through contact with Bent Hansen Sass (bhs@dmi.dk) and Roman Nuterman (nuterman@nbi.ku.dk). Documentation, educational materials and practical exercises are available from http://hirlam.org, http://hirlam.org/index.php/documentation/chemistry-branch and YSSS training schools: http://netfam.fmi.fi/YSSS08, http://www.ysss.osenu.org.ua and http://aveirosummerschool2014.web.ua.pt.

**Acknowledgements:**

This work was realised within and supported by the HIRLAM-A,-B,-C projects, COST Actions 715, 728 and ES1004 EuMetChem, and several European projects: EC FP5 ELCID, FUMAPEX, FP6 EnviroRISKS, FP7 MEGAPOLI, TRANSPHORM, MACC, PEGASOS and MarcoPolo; NordForsk projects - NetFAM, MUSCATEN, CarboNord, CRAICC-PEEX, CRUCIAL and others. Meteorological data were provided by the Paris measurement campaign of the FP7 EU MEGAPOLI project and by the Basque Meteorological Agency (EUSKALMET). The authors are greatly thankful to the colleagues involved into the model developments and applications at different stages: from the DMI team: J. Chenevez, A. Gross, K. Lindberg, P. Lauritzen, C. Petersen, X. Yang, L. Laursen, J.H. Sørensen, B. Amstrup, and from collaborators teams and PhD students: A. Mazeikis (Lithuania), S. Ivanov, Yu. Palamarchuk (Ukraine), S. Smyshlyaev, S. Mostamandy, E. Morozova, Yu. Gavrilova, A. Suhodskiy, A. Penenko (Russia), H. Toros (Turkey), K. Bostanbekov (Kazakhstan), A. Stysiak (Denmark). The authors are thankful to C. A. Skjøth (University of Worcester, UK) for providing the tree species inventory (TSI) data; European Forest Institute (EFI) - for broadleaved forest data; Danish Asthma Allergy Association - for birch pollen observation data; M. Sofiev and P. Siljamo (FMI, Helsinki) - for fruitful discussions of the birch pollen modelling issues. The handling topical editor Dr. Jason Williams, as well as Prof. Nicolas Moussiopoulos and two anonymous reviewers are thanked for thorough reviews, and for many valuable comments that substantially improved this article.

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

**Annex 2:** Abbreviations and acronyms used in this article:

| | |
|---|---|
| ACT | Atmospheric chemical transport |
| AHF | Anthropogenic heat flux |
| ALADIN | Aire Limitée (pour l') Adaptation dynamique (par un) Développement InterNational (model and consortium) |
| AOD | Aerosol Optical Depth |
| AQ | Air Quality |
| AQMEII | Air Quality Model Evaluation International Initiative |
| AROME | Application of Research to Operations at Mesoscale-model (Météo-France) ARW. The Advanced Research WRF solver (dynamical core) |
| BC | Black Carbon |
| BEP | Building Effect Parameterization |
| CAC | Chemistry-Aerosol-Cloud model (tropospheric box model) |
| CarboNord | Nordic project "Impact of Black Carbon on Air Quality and Climate in Northern Europe and Arctic" |
| CBM-IV | The modified implementation of the Carbon Bond Mechanism version IV |
| CBM-Z | CBM-Z extends the CBM-IV to include reactive long-lived species and their intermediates, isoprene chemistry, optional DMS chemistry |
| CHIMERE | A multi-scale CTM for air quality forecasting and simulation |
| CISL | Cell-integrated semi-Lagrangian (transport scheme) |
| COGCI | Copenhagen Global Climate Initiative |
| CORINE | European land-use database |
| COST | European Cooperation in Science and Technology (http://www.cost.eu/) |
| COSMO | Consortium for Small-Scale Modelling |
| COSMO-ART | COSMO + Aerosols and Reactive Trace gases |
| CPU | Central Processing Unit |
| CRAICC-PEEX | CRyosphere-Atmosphere Interactions in a Changing Arctic Climate - Pan Eurasian EXperiment |
| CRUCIAL | Nordic project "Critical steps in understanding land surface atmosphere interactions: from improved knowledge to socioeconomic solutions |
| CTM | Chemistry-Transport Model |
| CWF | Chemical Weather Forecasting |
| DMI | Danish Meteorological Institute |
| DMS | Dimethyl sulphide |
| DEHM | Danish Eulerian Hemispheric Model |
| ECMWF | European Centre of Medium-Range Weather Forecasts |
| EFI | European Forest Institute |
| ECHAM5-HAM | Global aerosol–climate model: Global GCM ECHAM (version 5) + Aerosol chemistry and microphysics package HAM (MPI for Meteorology, Hamburg) |
| ENCWF | European Network on Chemical Weather Forecasting |
| Enviro-HIRLAM | HIgh Resolution Limited Area Model HIRLAM with chemistry (DMI and collaborators) |
| EnviroRISKS | EU FP6 project: 'Environmental Risks: Monitoring, Management and Remediation of Man-made Changes in Siberia' |
| EPA | USA Environmental Protection Agency |
| ESM | Earth System Modelling |
| EuMetChem | The COST Action ES1004 – European framework for online integrated air quality and meteorology modelling (eumetchem.info) |
| ETEX | European Tracer Experiment |
| FAR | False alarm ratio |
| FMI | Finnish Meteorological Institute |
| FP5,6,7 | European Union Framework Programs |
| FUMAPEX | EU FP5 project "Integrated Systems for Forecasting Urban Meteorology, Air Pollution and Population Exposure" |
| GADS | Global Aerosol Data Set |
| GAW | Global Atmosphere Watch (WMO Programme) |
| GEOS-Chem | GEOS–Chem is a global 3-D chemical transport model (CTM) for atmospheric composition driven by meteorological input from the Goddard Earth Observing System (GEOS) of the NASA Global Modeling and Assimilation Office |
| GIS | Geographical Information System |
| GLCC | Global Land Cover Characterization |
| HAM | Simplified global primary aerosol mechanism model |
| HARMONIE | Hirlam Aladin Research on Meso-scale Operational NWP in Europe (model) |
| HCB | HIRLAM Chemical Branch |
| HIRLAM | HIgh Resolution Limited Area Model (http://hirlam.org/) |
| HR | Hit rate |
| IC/BC | Initial / Boundary Conditions |
| IFS | Integrated Forecast System (ECMWF) |
| ILMC | Posteriori iterative locally mass-conserving filter |
| ISBA | Interaction Soil- Biosphere- Atmosphere land surface scheme |
| IS4FIRES | Global biomass burning (wildfires) emission inventory developed by FMI |
| KPP | Kinetic Pre-Processors |
| LAI | Leaf Area Index |
| LMCSL | Locally Mass Conserving Semi-Lagrangian scheme |
| LUCY | Large scale Urban Consumption of energY model |
| LW | Long-wave radiation |
| MA | Model accuracy |
| M7 | Modal aerosol model |
| MACC | Monitoring Atmospheric Composition and Climate (EU project) |
| MEGAPOLI | EU FP7 project 'Megacities: Emissions, urban, regional and Global Atmospheric POLlution and climate effects, and Integrated tools for assessment and mitigation' (http://megapoli.info/) |
| MESO-NH | Non-hydrostatic mesoscale atmospheric model (French research community) |
| MetM | Meteorological prediction model |
| MOZART | Model for Ozone And Related Tracers (global CTM) |
| NetFAM | Nordic Research Network on Fine-scale Atmospheric Modelling |
| NMVOC | Non-Methane Volatile Organic Compounds |
| NWP | Numerical Weather Prediction |
| OC | Organic Carbon |
| ODE | Ordinary Differential Equation |
| OR | Odds ratio |
| PBL | Planetary Boundary Layer |
| POFD | Probability of false detection |
| OPAC | Optical Properties of Aerosols and Clouds (software library module) |
| PEGASOS | EU FP7 project: Pan-European Gas-Aerosol-Climate interaction study (http://pegasos.iceht.forth.gr/) |
| PM | Particulate Matter in two size bins – 2.5 µm and 10 µm ($PM_{2.5}$ and $PM_{10}$) |
| RBM | Radical balance method |
| RACM | Regional Atmospheric Chemistry Mechanism |
| SL | Semi-Lagrangian scheme |
| SMHI | Swedish Meteorological and Hydrological Institute |

| | |
|---|---|
| SM2-U | Soil Model for SubMeso - Urbanized version |
| STRACO | Soft TRAnsition and Condensation (Cloud scheme) |
| SW | Short Wave radiation |
| TKE-CBR | Turbulent Kinetic Energy Cuxart, Bougeault and Redelsperger scheme |
| TNO | the Netherlands Organisation for Applied Scientific Research |
| TRANSPHORM | EU FP7 project: 'Transport related Air Pollution and Health impacts - Integrated Methodologies for Assessing Particulate Matter' |
| TSI | Tree Species Inventory |
| TUV | Tropospheric Ultraviolet-Visible Model |
| TVA | Tennessee Valley Authority |
| UBL | Urban Boundary Layer |
| UHI | Urban Heat Island |
| YSSS | Young Scientist Summer School |
| VOC | Volatile Organic Compounds |
| WHO | World Health Organization |
| WMO | World Meteorological Organization |