# Peer review of "Enviro-HIRLAM online integrated meteorology-chemistry modelling system: strategy, methodology, developments and applications (v. 7.2)"

_Geoscientific Model Development, 2016_

## Short Comment (SC1) · 16 Jan 2017

Dear authors,

in my role as Executive editor of GMD, I would like to bring to your attention our Editorial version 1.1:

http://www.geosci-model-dev.net/8/3487/2015/gmd-8-3487-2015.html

This highlights some requirements of papers published in GMD, which is also available on the GMD website in the 'Manuscript Types' section:

[Figure]

http://www.geoscientific-model-development.net/submission/manuscript_types.html

In particular, please note that for your paper, the following requirement has not been met in the Discussions paper:

- "The main paper must give the model name and version number (or other unique identifier) in the title."

Please add a version number for your the ENVIRO-HIRLAM in the title upon your revised submission to GMD.

Yours,

Astrid Kerkweg

———————————————

---

## Referee Comment (RC1) · N. Moussiopoulos (Referee) · 4 Feb 2017

This manuscript provides a thorough presentation of Enviro-HIRLAM representing one of the first serious development efforts towards implementing a fully online coupled meteorological and chemical weather model. It contains detailed descriptions of methodology selected and implementation followed, including some coverage of less well-defined aspects of online coupling and performance evaluation. The paper is well written and contains a large amount of information. A section on model applications provides additional insight on the extremely important aspects of evaluation and validation.

As the overall assessment of the present referee, the paper successfully describes the remarkable effort that has been devoted to the development of a state-of-the-art online meteorological and chemical weather model. It is adequately referenced and contains detailed explanations of the main physical mechanisms and selected parameterisations. It also highlights some of the more promising aspects of the coupling idea, both in the area of aerosol-radiation treatment and in cloud microphysics.

The only weak point in the manuscript is the rather sketchy discussion of the extent to which the explicit introduction of all effects will lead to improvements in model performance. Section 4 of the manuscript represents of course an honest attempt to summarise what we know on the effect of coupling in model performance for different applications. The authors are encouraged to provide more explicit comments in this respect. This should be combined with a more thorough discussion on how all parameters required in the various process parameterisations could be fine-tuned (for instance, expanding the comments made in the last four lines of the paper).

In the below listed specific comments references are made to specific lines in the text.

1. Methodology and modelling system structure

a. The model coupling implements aerosol impacts on radiation (direct and semi-direct effects) and on clouds (first and second indirect effects), l. 110-111. It appears appropriate to include an explicit reference to COST action ES1004 in the framework of which these effects were extensively discussed.

b. The cloud feedback module includes some rather advanced approximations (l. 287-293); the reader would welcome more remarks on the extent to which this complex cloud model has been validated.

c. The present HIRLAM NWP model core is based on the hydrostatic approximation (l. 127) which could be a serious limitation over complex terrain (l. 508) and/or in cases of

nesting down to urban areas. A plan for a transition to a new, non-hydrostatic platform (e.g. HARMONIE, l. 135) is mentioned, but more information in this respect would be helpful.

d. The atmospheric chemistry modules implement a wide array of new parameterisations and numerical schemes (page 4, l. 141-192). Although these were obviously validated separately, their combined implementation in a coupled model definitely needs further validation. Did the authors take already actions in this direction, and if not, what are their plans?

e. The aerosol dynamics model introduces a very interesting classification of particles depending both on particle size and particle composition per emission source. This could allow, in theory, a separate per-source type treatment of particles throughout the chemical mechanism. But is there such a procedure (with potential applications in source apportionment) really implemented or planned?

f. Specific emission models for anthropogenic biomass burning (e.g. wildfires) are included (section 2.5). These are based on satellite or other inventory-estimates of yearly fluxes that are temporally disaggregated using pre-defined temporal profiles. Are the latter site dependent, and which is the origin of the coefficients used by the authors?

g. The model contains several "urbanisation" features (section 2.7), including a subset of previously proposed urban parameterisations (Martilli, Dupont, Masson, Grimmond et al.). This is an interesting and original approach, but there are several concerns on how it is implemented (please see comment 2c below).

h. In l. 372-378 some aspects of the so-called locally mass conserving semi-Lagrangian (LMCSL) transport scheme are described. The description emphasizes the approximate mass-conserving properties of the algorithm for 1st neighbour cells, but one could ask whether and how is mass consistency ensured in the larger scale. In l. 389-390 it is stated "[. . .]Enviro-HIRLAM is not formally wind-mass consistent regarding tracer transport". The authors should discuss possible consequences of this failure.

2. Model applications and validation

a. Sensitivity studies on the model response to aerosol effects do indicate some strong "signals" (difference between coupled and uncoupled runs), e.g. l. 418. But these do not necessarily imply an improved model performance, and the authors should state this clearly in the manuscript, cf. l. 420 "[Korsholm (2009)] found a marginally improved agreement[...]", and l. 464-467 "However...it is too early to make conclusions about the improvement of precipitation forecasting by implementation of the indirect aerosol effects, because of large uncertainties in parameterisation . . . and due to adjustments of such effects. . .and constants".

b. This referee believes that careful tuning is needed in view of the large number of parameters in the complex feedback modules, especially with regard to cloud effects. It is not obvious how and to what extent this could be achieved only by comparing final simulation results (i.e., without a further quantitative study of the cloud physical mechanisms themselves).

c. An evaluation application for the urbanisation modules was performed for the cities of Paris and Bilbao. There are several issues regarding this application that are neither explained in the text nor in the referenced publications:

i. A domain spatial resolution of 2.5 km appears to be insufficient for such an application.

ii. The resolution of the BEP dense sub-grid is not mentioned. Is it also 2.5 km?

iii. The authors seem having assumed only four urban classes, cf. Figure 10. Such a classification would ignore the important role of green urban areas in UHI evolution.

iv. Is the 2.5×2.5 arc-minute resolution (∼5km) of the AHF data adequate for assessing UHI effects in an urban scale? In the Bilbao case it appears that the entire urban area

Interactive
comment

is covered (and classified) in only 16 cells!

v. Are AHF data constant during the day, or do the authors assume an intrinsic diurnal profile?

vi. Values of 40 or 60 Wm-2 for the AHF are mentioned. Is this a mean annual value or a daily estimation following a seasonal profile?

vii. Concerning the validation process, it is unclear whether a combination of statistical indicators is used or just the correlation coefficient. Not much evidence is presented (e.g. in form of figures or tables) that the model reproduces satisfactorily the mesoscale features.

viii.It is well documented in the literature that the Paris UHI is expanding just after midnight, but not that this expansion lasts until 11 UTC, especially during a summer period. Comments by the authors would be welcome.

ix. Confusion is caused by the fact that in the second paragraph of section 3.2 the authors claim that the model was applied for July 2009, while in the last paragraph of the same section they write ". . .showed that under calm conditions during summer and winter. . .".

d. Enviro-HIRLAM is operationally used for birch pollen forecasting in Denmark. This appears to be one of the more mature applications of the model, with rather advanced emission, deposition and scavenging modules. However, no mention is made on the effect of online coupling (and the relevant feedbacks) on these simulations. In the conclusions it is mentioned that feedbacks are not important in pollen forecast (l. 711-712). How did the authors reach this conclusion?

e. Section 3.4 attempts an evaluation of the feedback effects on air pollution forecasting. It is mentioned that online coupling improves the forecast skill, however without referring to specific applications, as for instance the MEGAPOLI Paris campaign.

From a technical point of view, the paper is excellent. Yet, the authors should check it

again for inconsistencies (e.g., both "online" and "on-line" are found in the manuscript).

---

## Referee Comment (RC2) · Anonymous Referee #2 · 6 Feb 2017

General comments:

It would appear that the primary objectives of the presented manuscript were to introduce, document and promote a 'fit-for-purpose' application of the Enviro-HILRAM model.

The Eviro-HILRAM model is well established in the community. It is being used and developed trough a broad international collaboration. It is important that a proper reference to this valuable tool is provided. The Authors made an effort to present the origin and evolution of the model over the years. Also, a short description of model

components and applications was provided. Specific comments and suggestions are given in the next section.

In the manuscript, the Authors advance terms and concepts of "online coupling", "fully online integration", "seamless meteorology-chemistry modelling", "two-way interacting", "on-line integration". The use of these terms is not consistent and confusing. Also, the concept of a meteorological/NWP model with chemistry was proposed, implemented and published earlier than the provided reference to Grell at al. (2005). Coupled chemistry-climate models were developed and used in the 1990s, cf. Steil et al., 2003 (doi:10.1029/2002JD002971), Austin and Butchart, 2003 (Q. J. R. Meteorol. Soc., 129, 3225–3249), de Grandpré et al., 2000 (J. Geophys. Res, 105, 26,475–26,492), among other publications. Thus, a proper historical and scientific perspective should be preserved, especially in a paper that presents "strategy and methodology" and dedicates several paragraphs to model evolution and origin.

The Authors introduced the term "biological weather". It is the understanding of the reviewer that this term refers to birch pollen modelling. However, the meaning of the term is unclear and probably misleading.

It is not evident, from the presented model description that it is a multiscale or a wideband atmospheric model. In most of the presented examples, the model domain covers the European continent. Application of the model to urban scale with a resolution of 2.5 km in a hydrostatic mode is rather problematic. The Authors should further comment and justify its use at the said resolution (cf. Lines 508-509).

The Authors provided references to all model components and applications. However, this paper should explicitly provide all 'vital model information' such as vertical structure, horizontal resolutions (with clearly stated limitations), numerical methods and approximations employed in different modules (components), modularity and scalability of components, examples of integration time and computer topology used for benchmarking. What is the required computer power, maximum number of computational

cores, can the model be run on a heterogeneous architecture with GPUs? All these characteristics should be addressed and tabulated with appropriate references and notes.

In several sub-sections, the Authors included a description of earlier versions of the model. Thus, it is not clear to the reader which parameterizations are used in the current version of the Enviro-HIRLAM model. It would be advisable to move these paragraphs to an appendix presenting development stages and perspective of the Enviro-HIRLAM model.

In Section 3 (Modelling system applications), the Authors refer to several earlier publications. It is not clear if the presented manuscript contains any results that were not published. It would be advisable to add a table (in Section 3) with a list of presented experiments and model versions used for simulations together with appropriate references. Also, if a figure is adopted from an earlier publication, a proper reference should be included in a figure caption.

Pollen module description should be moved from Section 3.3 (Pollen forecast) to Section 2 (system description).

Sub-section 3.4 should be moved and inserted as 3.1

Overall, the justification of advantages of the on-line approach is not sufficiently demonstrated. Verification aspects should be included in a more coherent way. Presented experiments refer to relatively short periods (one summer month). Results for the gas phase chemistry are not discussed.

The Authors should restructure the manuscript to emphasise the overall modelling philosophy and future directions of the proposed model development and applications.

Specific comments:

The presented comments are in a sequential order and refer to the line numbering in the presented manuscript.

L22: "Online integrated passive pollutant transport" - the same term should not be used for the simplified approach.

L27: What is "effective chemistry"?

L35-36: The section title is too long and awkward.

L68: The style of Figure 1 does not conform to a convention used in scientific publications.

L108: "current new version" - should be either "current" or "new"?

L128: "main meteorological fields" - please define.

L142: How long are the "long-term runs". Please explain and justify.

L175-185: The whole section on photolysis rates is confusing and misleading.

L177: Please explain how the ozone column is set above the model top.

L181: The assertion that the 8-stream method is "the most accurate" should be justified.

L282: In Figure 4 X-axes have different units.

L343: What is "traditional" SL? Please provide a reference.

L382: Figure 6: The presented figure alone does not prove that the model can deal with sharp gradients.

Line 389: What is TR4?

Line 390: The mental jump referring to "formal conservation" should be explained.

L407: The title is confusing, and the whole section is too long. Half of the first paragraph refers to urban applications, which are discussed in the next section.

L497: It is wrong to assert that higher correlation implies that the model is "closer to

observations."

L505: The ability of a weather prediction model (i.e. HIRLAM) to reproduce meso-scale processes at the regional scale should not depend on the use of an urban parameterization. The presented conclusions do not belong in Section 3.2.

L654: The calculations were analysed for one month (July 2010) only. Thus, the sentence is too general.

L656: "crude model resolution" - what does it mean?

The use of the English language:

The Authors should pay particular attention to the use of articles, prepositions and tenses in the revised manuscript. Also, the Authors used words that do not exist i.e. Line 255: 'to split' is an irregular verb – the simple past tense is 'split', or words in a wrong context i.e. Line 187 'Heterogenic chemistry' should be 'Heterogeneous chemistry'.

Recommendation:

In the opinion of this reviewer, the presented manuscript could constitute an important contribution documenting the Enviro-HILRAM model. The paper should be published after major revisions.

---

## Referee Comment (RC3) · Anonymous Referee #3 · 9 Feb 2017

The manuscript presents the online coupled model Eviro-HIRLAM, which is well known in the atmospheric modelers community. The manuscript is well structured and provides a comprehensive presentation of Enviro-HIRLAM development with a description of the different approaches and physical schemes implemented during the model evolution. The computational schemes and parameterizations adopted by the models are properly introduced and referenced. A minor shortcoming of this approach is that it is somewhere not very clear to the reader which computational scheme is the one chosen for the present version of the model or what alternatives are provided to the user.

[Figure]

A relevant number of applications are referenced for almost all the model development fields. Some of the items (e.g. pollen) are described providing explicit summary of the overall results that make the paper more readable and useful for a reader that is not willing to read the large number of referenced papers and documents. Other examples of application are mainly discussed through references and do not allow the reader to appreciate the model effectiveness and the improvement offered by the online modelling approach.

If the general approach of online coupling is physically sound and it can be agreed that it will probably become the prevailing modelling approach in the next future, the manuscript does not clarify, through its application examples, to what extent the online coupling and the main parameterizations introduced (e.g. urbanization) provide an improvement of model capability to predict observed pollutant concentrations and key meteorological parameters. An improvement of the analysis of the online coupling effectiveness is desirable and would make the manuscript more complete, interesting and valuable.

Text and figures include a large number of acronyms for project names, parameterization schemes, etc. Even if many of them are known, it is quite difficult for the reader to know and remind all their meaning. It would be helpful to add an acronym legend section.

Specific comments:

Section 1. Methodology

Lines 72-75 The authors say that Enviro-HIRLAM is being used for different research project, but most cited project have already concluded they activity. In the Figure 1 lowest box most project mentioned as ongoing are finished since a few years.

Section 2.1 Modelling system structure

Line 92 The URL http://hirlam.org/trac/wiki/ is password protected and therefore not

accessible to the reader. It should be substituted with an open access web site.

Section 2.3 Atmospheric chemistry

It is not clear if the "tropospheric sulfur cycle" is a simple scheme alternative to the CBM-Z, that is presently maintained for simplified simulations (what is the specific interest?), or if it is an obsolete option which is going to be abandoned. It is not specified how the CBM-Z gas-phase chemistry scheme is interfaced with the M7 aerosol module. Due to the relevance of secondary particle production modelling, more details would be appreciable to provide a comprehensive model description.

Lines 171-172 The authors say they "use KPP tools to create the gas-phase chemical mechanisms including the solvers for three chemical mechanisms." What are the three mentioned chemical mechanisms? Only two of them have been previously presented: a) Tropospheric Sulfur Cycle, b) Gas-phase chemistry (CBM-Z).

Lines 172-173 The authors say that Rosenbrock solver is usually selected. Why?

Line 190 What is "NWP-Chem-Liquid"?

Section 2.4. Aerosol formation, dynamics and deposition

Line 197 Is CAC still available in Enviro-HIRLAM or it is mentioned only for historical development reasons?

Lines 205-206 Is the aerosol type identity maintained through the model simulation and provided as separated output contribution to the total PM?

Section 2.5. Emission modules and pre-processor

Line 254 Does wildfires emission module consider PM only or gas phase pollutants too?

Line 274 What are "transported modes"?

Section 2.7. Urban parameterizations and models urbanization

This section is relevant because it highlights the need for a mass conserving transport scheme in on-line coupled NWP and ACT models. For offline coupling this request is less strict because mass consistency is usually guaranteed by the coupler module.

Line 311 Bracket missing.

Line 312 Grid nesting is an effective technique to increase model resolution but it is rather confusing to consider it a method to represent urban areas.

Line 315 The "calculation of the urban mixing height based on prognostic approaches" is neither described nor commented in the following text.

Section 2.8. Transport schemes

Line 371 Is hat symbol missing on "modified weight" in equation 6 ?

Line 377 ""is are" should be corrected

Lines 388-390 This sentence concerning Enviro-HIRLAM mass consistency for tracer transport should be better explained and discussed. What are the possible limitations caused by this lack of mass conservation? What is TR4?

Section 3 Modelling system applications

What are the mentioned "EnvCLIMA, Enviro-HIRHAM"?

Lines 415-418 Do the mentioned temperature changes due to indirect effects improve model results? How relevant is the improvement? The reference given by the authors is to a Project report that can be hardly available, not to a journal publication. In the following sentence (lines 420-421) the authors mention a marginal improvement on surface temperature. They also mention a redistribution effect on NO2 concentration, but they do not specify is this effect improves model results.

Lines 442-444 and Figure 9 The authors say "the ENV run bias for precipitation with respect to its frequency and amount has been decreased compared to the REF model

run (Fig. 9)." Legends printed on the pictures seem opposite to what indicated in the caption (Enviro-HIRLAM on the left). Results showed in Figure 9 seem different during different parts of simulation: until July 21st the right side simulation seems better, while the left side one seems better during the last part of the simulation. What is the difference of the overall biases?

Lines 480-489 A grid size of 2.5 km seems quite crude to resolve Bilbao city. In x and y directions the city seems to be described by 2 to 4 grid cells which can be hardly considered sufficient to develop a "urban signal". Why has not been used a finer resolution? Is it due to the hydrostatic model limitations?

Figure 10 Why different land use classifications have been used for the two considered cities? What is the P01 modelling domain mentioned in the caption?

Line 498 Does 10% improvement refer to the correlation value?

Lines 499-500 It is not clear how the mentioned correlations have been computed. Time correlation for separated hours? How many stations have been used to compute the mentioned correlations?

Lines 501-504 Where the mentioned results for Bilbao better than those obtained without urbanization? Was the improvement significant?

Lines 512-535 The authors show that model urbanization allows to describe UHI phenomenology in Paris and Bilbao, but they do not discuss if the urbanization improves results and reduces possible model bias with respect to urban observations.

Lines 635-639 The mentioned effects of aerosol feedbacks on chemical composition are quite interesting. Did the mentioned changes on NO2 and O3 improve model results and increase its capability to reproduce measured values?

Figure 15 Right side color scale legend needs correction. How are correlations for separated hours computed?

Lines 675-677 The authors mention new model applications without providing any detail about recent results potentially relevant and interesting for the readers. The mentioned feedback mechanisms evaluation is one of the key point of the paper.

Section 4 Conclusions

Lines 692-702 These sentences contain repetitions of the same concepts that could be removed.

---

## Author Comment (AC2) · 31 Mar 2017

**Authors response to comments of the Referee #1 Prof. N. Moussiopoulos**

**We thank the Referee #1 Prof. N. Moussiopoulos for the interesting and important comments on our manuscript. All the individual comments are addressed below in red.**

This manuscript provides a thorough presentation of Enviro-HIRLAM representing one of the first serious development efforts towards implementing a fully online coupled meteorological and chemical weather model. It contains detailed descriptions of methodology selected and implementation followed, including some coverage of less welldefined aspects of online coupling and performance evaluation. The paper is well written and contains a large amount of information. A section on model applications provides additional insight on the extremely important aspects of evaluation and validation.
As the overall assessment of the present referee, the paper successfully describes the remarkable effort that has been devoted to the development of a state-of-the-art online meteorological and chemical weather model. It is adequately referenced and contains detailed explanations of the main physical mechanisms and selected parameterisations. It also highlights some of the more promising aspects of the coupling idea, both in the area of aerosol-radiation treatment and in cloud microphysics.
The only weak point in the manuscript is the rather sketchy discussion of the extent to which the explicit introduction of all effects will lead to improvements in model performance.

**Response:**
The paper is focusing mostly on the model description and its applications, therefore it was not much space for detailed discussion of specific effects of different model improvements. Some aspects were published in previous papers, some are still in new specific papers to be submitted (e.g. for pollen applications, operational air quality forecasting).
We have extended this part in the revised version.

Section 4 of the manuscript represents of course an honest attempt to summarise what we know on the effect of coupling in model performance for different applications. The authors are encouraged to provide more explicit comments in this respect. This should be combined with a more thorough discussion on how all parameters required in the various process parameterisations could be fine-tuned (for instance, expanding the comments made in the last four lines of the paper).

**Response:** The Section 4 is extended correspondingly in the revised version.

In the below listed specific comments references are made to specific lines in the text.
1. Methodology and modelling system structure
a. The model coupling implements aerosol impacts on radiation (direct and semidirect effects) and on clouds (first and second indirect effects), l. 110-111. It appears appropriate to include an explicit reference to COST action ES1004 in the framework of which these effects were extensively discussed.

**Response**: Thanks, agree. The corresponding reference is included.

b. The cloud feedback module includes some rather advanced approximations (l. 287-293); the reader would welcome more remarks on the extent to which this complex cloud model has been validated.

**Response**:

Abdul-Razzak and Ghan parameterization for aerosol activation has been extensively tested in many online-coupled weather and climate models. However, the STRACO cloud microphysics scheme with parameterizations of aerosol activation, cloud droplets nucleation, sedimentation, evaporation, self-collection, has never been thoroughly evaluated, only with 1D column HIRLAM. These evaluation results are not ready to be published yet, but will be analysed and published on further steps.

c. The present HIRLAM NWP model core is based on the hydrostatic approximation (l. 127) which could be a serious limitation over complex terrain (l. 508) and/or in cases of nesting down to urban areas. A plan for a transition to a new, non-hydrostatic platform (e.g. HARMONIE, l. 135) is mentioned, but more information in this respect would be helpful.

**Response:**
Yes, we agree about the limitations and write openly about them. The new version under HARMONIE is only under development and only some elements are realised so far, so it is too early to describe it extensively it in more details at this stage.
The non-hydrostatic HARMONIE-AROME model includes only some aerosol effects. The physics included in this version of HARMONIE has recently been detailed by Bengtsson et al. (2017). HARMONIE-AROME is based partly on Meso-NH (Mesoscale Non-Hydrostatic atmospheric model), which is a cloud resolving model that includes state-of-the-art chemistry and aerosol interactions (e.g. Berger et al. 2016). Meso-NH can, however, not be run as a near real time NWP model, which is possible with Enviro-HIRLAM.
Corresponding extension text is included in the revised version.
The following additional references are included:
Bengtsson, L., U. Andrae, T. Aspelien, Y. Batrak, J. Calvo, W. de Rooy, E. Gleeson, B. Hansen-Sass, M. Homleid, M. Hortal, K. Ivarsson, G. Lenderink, S. Niemelä, K. P. Nielsen, J. Onvlee, L. Rontu, P. Samuelsson, D. Santos Muñoz, A. Subias, S. Tijm, V. Toll, X. Yang, and M. Ødegaard Køltzow, 2017: The HARMONIE-AROME model configuration in the ALADIN-HIRLAM NWP system. Mon. Wea. Rev. doi:10.1175/MWR-D-16-0417.1, in press.
Berger A., M. Leriche, L. Deguillaume, C. Mari, P. Tulet, D. Gazen and J. Escobar, 2016: Modeling Formation of SOA from Cloud Chemistry with the Meso-NH Model: Sensitivity Studies of Cloud Events Formed at the Puy de Dôme Station. In: Steyn D., Chaumerliac N. (eds) Air Pollution Modeling and its Application XXIV. Springer Proceedings in Complexity. Springer, Cham

d. The atmospheric chemistry modules implement a wide array of new parameterisations and numerical schemes (page 4, l. 141-192). Although these were obviously validated separately, their combined implementation in a coupled model definitely needs further validation. Did the authors take already actions in this direction, and if not, what are their plans?

**Response:**
Yes, these chemical schemes/solvers were tested and validated as standalone versions (Reference: Shalaby, A., Zakey, A. S., F., Giorgi, and M.M. AbdelWahab "Coupling of Regional Climate Chem Aerosol Model", Ph.D. thesis, Faculty of Science, Cairo University-Egypt, 2012). Six environmental/smog chamber experiments were used to validate the gas-phase schemes and different chemical solvers as box models.
The Tennessee Valley Authority (TVA) and the EPA chamber experiments were used to evaluate the different gas-phase schemes and different chemical solvers.  Namely, TVA005 and TVA006 are designed to test the simple system of NOx; TVA068 is designed to test a simple mixture of VOC with very high NOx. EPA069A, EPA073A and EPA150A are used to validate the schemes with low NOx concentration and high VOC concentration.
Also, the same chemistry schemes/solvers are coupled with the Regional Climate Model (Reference: Shalaby, A., Zakey, A. S., Tawfik, A. B., Solmon, F., Giorgi, F., Stordal, F., Sillman, S., Zaveri, R. A., and Steiner, A. L.: Implementation and evaluation of online gas-phase chemistry within a regional

climate model (RegCM-CHEM4), Geosci. Model Dev., 5, 741-760, doi:10.5194/gmd-5-741-2012, 2012.)

However such validations need to be further continued and completed, this is the issue for further analysis.

Corresponding extension text and the above mentioned additional references are included in the revised version.

e. The aerosol dynamics model introduces a very interesting classification of particles depending both on particle size and particle composition per emission source. This could allow, in theory, a separate per-source type treatment of particles throughout the chemical mechanism. But is there such a procedure (with potential applications in source apportionment) really implemented or planned?

**Response**:
The particles classification with respect to their size and composition is based on aerosols classification in M7 microphysics module. As for total particulate matter the splitting is in species, the procedure follows guidelines and recommendations associated with different emission inventories. For now, there is no procedure for source apportionment in the model or plans to do that in the nearest future.

f. Specific emission models for anthropogenic biomass burning (e.g. wildfires) are included (section 2.5). These are based on satellite or other inventory-estimates of yearly fluxes that are temporally disaggregated using pre-defined temporal profiles. Are the latter site dependent, and which is the origin of the coefficients used by the authors?

**Response**:
The Finish Meteorological Institute developed the IS4FIRES (http://is4fires.fmi.fi) wildfires emission inventory. The IS4FIRES inventory provides temporal profiles for emissions disaggregation with site dependency. However, the profiles used in Enviro-HIRLAM runs for this paper are site independent (mean) and are the functions of a local time only. They were also provided by FMI for AQMEII-2 (COST Action ES1004) initiative.
**Changes in manuscript:**
**L257:** The biomass burning emissions typically show a diurnal cycle variability, and therefore, corresponding coefficients are applied (*Giglio, 2007*).
**Added to reference list:** Giglio, L., 2007: Characterization of the tropical diurnal fire cycle using VIRS and MODIS observations, Remote Sensing of Environment, 108, 4, pp. 407-421.

g. The model contains several "urbanisation" features (section 2.7), including a subset of previously proposed urban parameterisations (Martilli, Dupont, Masson, Grimmond et al.). This is an interesting and original approach, but there are several concerns on how it is implemented (please see comment 2c below).

**Response:**
The details of implementations of different urban modules, our own developments and comparisons of different approaches and modules were published in previous papers (Mahura et al., 2005b, 2006a, 2007a, 2008bc; Baklanov et al., 2005, 2008), so we don't describe them in this paper. The main approach includes an integration of the urban modules into the ISBA (Interaction Soil- Biosphere-Atmosphere) land surface scheme of the NWP / HIRLAM model. The urban modules are activated only on those grid cells of the model domain where the urban fraction is presented.
More explanations and corresponding references on the above papers are included in the revised version.

h. In l. 372-378 some aspects of the so-called locally mass conserving semi-Lagrangian (LMCSL) transport scheme are described. The description emphasizes

the approximate mass-conserving properties of the algorithm for 1st neighbour cells, but one could ask whether and how is mass consistency ensured in the larger scale. In l. 389-390 it is stated "[: : :]Enviro-HIRLAM is not formally wind-mass consistent regarding tracer transport". The authors should discuss possible consequences of this failure.

**Response:**
We have added a sentence to clarify that mass-wind inconsistency is a minor problem. The traditional HIRLAM is (at least in principle) wind-mass consistent. In Enviro-HIRLAM where all moisture fields are transported with the LMCSL scheme, there is no formal consistency, yet, since precipitation is very similar to that in HIRLAM (except for individual convective systems that are chaotic/unpredictable in their nature), the mass-wind inconsistency is small in practice.

**Suggested new version of lines 372-378 in the original text (changes in bold)**
As the traditional SL scheme, the LMCSL is not **inherently** monotonic or positive definite. Therefore a posteriori iterative locally mass-conserving (ILMC) filter was developed, Sørensen et al. (2013).
……
It should be noted that the dynamical core in Enviro-HIRLAM is identical to that of HIRLAM. Thus, the dry-air density for dynamics is calculated using a traditional SL approximation to (4), i.e. not the LMCSL. Therefore, the Enviro-HIRLAM is not formally wind-mass consistent regarding tracer transport. **However, the large scale precipitation fields in the traditional HIRLAM and Enviro-HIRLAM are very similar (see, e.g., Figure 4 in Sørensen et al. (2013)), which suggests that wind-mass inconsistency is of minor importance.**

2. Model applications and validation
a. Sensitivity studies on the model response to aerosol effects do indicate some strong "signals" (difference between coupled and uncoupled runs), e.g. l. 418. But these do not necessarily imply an improved model performance, and the authors should state this clearly in the manuscript, cf. l. 420 "[Korsholm (2009)] found a marginally improved agreement[: : :]", and l. 464-467 "However: : :it is too early to make conclusions about the improvement of precipitation forecasting by implementation of the indirect aerosol effects, because of large uncertainties in parameterisation : : : and due to adjustments of such effects: : :and constants".

**Response:**
Yes, we agree with the reviewer but don't see contradictions between these statements. Sensitivity studies on the model response to aerosol effects indicate strong "signals", but it doesn't guaranty improvements. E.g., Korsholm (2009) considered evaluations only for some elements (e.g., the coupling interval) in the previous analysis and made corresponding conclusions about the improvements. Other feedback mechanisms effects, especially for aerosol-cloud interactions, studied mostly as sensitivity studies or evaluations for short-term case studies.
The model formulations have only been tested on a case basis and although strong signals have been found this does not imply improved meteorological performance of the model. In particular, testing over longer periods including all seasons was not conducted that time. Furthermore, the interactions between aerosols and the cloud ice-phase are not in a state where improvements would be expected. Therefore we wrote that conclusions about the improvement of precipitation cannot be done at this stage and need more analysis.
Recently similar evaluation studies are realised within the CarboNord project for monthly and annual validation studies. However, they are recently started.

b. This referee believes that careful tuning is needed in view of the large number of parameters in the complex feedback modules, especially with regard to cloud effects. It is not obvious how and to what extent this could be achieved only by comparing final simulation results (i.e., without a further quantitative study of the cloud physical mechanisms themselves).

**Response:**
We fully agree with the reviewer. The STRACO cloud scheme contains fairly simplified cloud microphysics (heavily parameterized). Hence, tuning is essential for the overall performance of the model, when it comes to precipitation and cloud physical properties. Further work to improve aerosol-cloud interaction and precipitation forecast is needed.

c. An evaluation application for the urbanisation modules was performed for the cities of Paris and Bilbao. There are several issues regarding this application that are neither explained in the text nor in the referenced publications:
i. A domain spatial resolution of 2.5 km appears to be insufficient for such an application.

**Response:**
Sensitivity tests demonstrated that the 2.5 km was the optimal resolution allowing at the same time to obtain satisfactory reproducibility of the large scale processes and to explore the urban effects at local scale without being diminished due to a coarse resolution, taking into account the limitations of the hydrostaticity of the NWP model.

ii. The resolution of the BEP dense sub-grid is not mentioned. Is it also 2.5 km?

**Response:** Yes, the BEP is also computed at 2.5 km resolution.

iii. The authors seem having assumed only four urban classes, cf. Figure 10. Such a classification would ignore the important role of green urban areas in UHI evolution.

**Response:**
Although we assumed four types of urban areas, the urban grid is not fully covered by urbanisation; it also contains a fraction of the green area, defined in the CORINE 2000. The classification and the percentage of urban/non-urban grid can be found in Gonzalez-Aparicio et al. (2010), page 17 Table 4.

iv. Is the 2.52.5 arc-minute resolution (5km) of the AHF data adequate for assessing UHI effects in an urban scale? In the Bilbao case it appears that the entire urban area is covered (and classified) in only 16 cells!
v. Are AHF data constant during the day, or do the authors assume an intrinsic diurnal profile?
vi. Values of 40 or 60 Wm-2 for the AHF are mentioned. Is this a mean annual value or a daily estimation following a seasonal profile?

**Response (for iv-vi):**
For the Bilbao study, Enviro-HIRLAM didn't implement any AHF parameterization and therefore, the AHF factors were estimated from the LUCY model, as a value for summer and winter without including any daily profile. The value was constant for the urban fraction in the 16 grid cells (it is multiplied – e.g. depends on urban fraction: if 100% -> then max value) covering the area of Bilbao (92 $km^2$). Although it is not as big as the Paris metropolitan area, the effects of the AHF and the UHI on the atmospheric boundary layer could be visible. A sensitivity analysis of the effects of the AHF and the UHI on the atmospheric boundary layer can be found in Gonzalez-Aparicio et al. (2014).

vii. Concerning the validation process, it is unclear whether a combination of statistical indicators is used or just the correlation coefficient. Not much evidence is presented (e.g. in form of figures or tables) that the model reproduces satisfactorily the mesoscale features.

**Response:**
The full validation process can be found in Gonzalez-Aparicio et al. (2013) and Gonzalez-Aparicio et al. (2014). The text summarised the overall performance over the two episodes analysed.

viii.It is well documented in the literature that the Paris UHI is expanding just after midnight, but not that this expansion lasts until 11 UTC, especially during a summer period. Comments by the authors would be welcome.

**Response:**
We agree that the Paris UHI is generally expanding just after the midnight and this is very well documented. In this paper we present the evolution of the UHI on the single day of the 28[th] July 2009. The UHI was expanding after midnight and the effect was visible up to 11 UTC, not meaning that the expansion lasted until 11 UTC.

ix. Confusion is caused by the fact that in the second paragraph of section 3.2 the authors claim that the model was applied for July 2009, while in the last paragraph of the same section they write ": : :showed that under calm conditions during summer and winter: : :".

**Response:**
The analysis described in the second and third paragraph of the section refers to July 2009, as the text describes. The last paragraphs indicate the outcomes presented in Gonzalez-Aparicio et al. (2013) and Gonzalez-Aparicio et al. (2014) and focused on winter and summer episodes.

d. Enviro-HIRLAM is operationally used for birch pollen forecasting in Denmark. This appears to be one of the more mature applications of the model, with rather advanced emission, deposition and scavenging modules. However, no mention is made on the effect of online coupling (and the relevant feedbacks) on these simulations. In the conclusions it is mentioned that feedbacks are not important in pollen forecast (l. 711-712). How did the authors reach this conclusion?

**Response:**
The current version of the birch pollen model presented in the paper has not been used in operational mode yet. Online coupling is important for birch pollen simulations due to dependency of the birch pollen emissions on meteorology. It is also specified in lines 550-552 of the paper.
The current version of Enviro-HIRLAM considers birch pollen as a passive tracer with no pollen feedbacks on meteorology. Online coupling (i.e. impact of meteorology on the emissions) is of main importance in the birch pollen study.

e. Section 3.4 attempts an evaluation of the feedback effects on air pollution forecasting. It is mentioned that online coupling improves the forecast skill, however without referring to specific applications, as for instance the MEGAPOLI Paris campaign.

**Response:**
As we mentioned, the considered evaluations were done only for some elements (e.g., the coupling interval) in the previous analysis and main conclusions about the improvements were done just for them. Other feedback mechanisms effects, especially for aerosol-cloud interactions, analysed mostly as sensitivity studies or evaluations for short-term episodes. Unfortunately, during the MEGAPOLI Paris measurement campaign we were not able to include measurement studies of aerosol-cloud interactions, so it was not possible to make evaluations of aerosol feedbacks vs the MEGAPOLI Paris data. So, we wrote only about a general reasonable performance of the model vs. measurement data. Corresponding corrections and explanations are made in the revised version of the paper.

From a technical point of view, the paper is excellent. Yet, the authors should check it again for inconsistencies (e.g., both "online" and "on-line" are found in the manuscript).

**Response:** Thanks a lot. It is corrected in the revised version.

---

## Author Comment (AC3) · 31 Mar 2017

**Authors response to comments of the Referee #2**

We thank the Anonymous Referee #2 for the interesting and important comments on our manuscript. All the individual comments are addressed below in red.

General comments:
It would appear that the primary objectives of the presented manuscript were to introduce, document and promote a 'fit-for-purpose' application of the Enviro-HILRAM model.
The Eviro-HILRAM model is well established in the community. It is being used and developed trough a broad international collaboration. It is important that a proper reference to this valuable tool is provided. The Authors made an effort to present the origin and evolution of the model over the years. Also, a short description of model components and applications was provided. Specific comments and suggestions are given in the next section.
In the manuscript, the Authors advance terms and concepts of "online coupling", "fully online integration", "seamless meteorology-chemistry modelling", "two-way interacting", "on-line integration". The use of these terms is not consistent and confusing.

**Response:** Thanks. The terminology is harmonised/corrected in the revised version.

Also, the concept of a meteorological/NWP model with chemistry was proposed, implemented and published earlier than the provided reference to Grell at al. (2005).
Coupled chemistry-climate models were developed and used in the 1990s, cf. Steil et al., 2003 (doi:10.1029/2002JD002971), Austin and Butchart, 2003 (Q. J. R. Meteorol. Soc., 129, 3225–3249), de Grandpré et al., 2000 (J. Geophys. Res, 105, 26,475–26,492), among other publications. Thus, a proper historical and scientific perspective should be preserved, especially in a paper that presents "strategy and methodology" and dedicates several paragraphs to model evolution and origin.

**Response:**
Thank you for the comments. The references are included in the revised version. However, more comprehensive historical overviews of coupled chemistry-meteorology models were done e.g. by Zhang (2008), Kukkonen et al. (2011), Baklanov et al. (2014).

The Authors introduced the term "biological weather". It is the understanding of the reviewer that this term refers to birch pollen modelling. However, the meaning of the term is unclear and probably misleading.

**Response:**
The "biological weather" term is defined in Klein et al. 2012 as "the short-term state and variation of concentrations of bioaerosols".
Thus in the current paper, biological weather refers to birch pollen modelling.
The reference to Klein et al. 2012 is included for clarification.

It is not evident, from the presented model description that it is a multiscale or a wideband atmospheric model. In most of the presented examples, the model domain covers the European continent. Application of the model to urban scale with a resolution of 2.5 km in a hydrostatic mode is rather problematic. The Authors should further comment and justify its use at the said resolution (cf. Lines 508-509).

**Response:**

Yes, the hydrostatic approximation of the model was a limitation to increase the resolution to perform the urban simulations. However, sensitivity tests demonstrated that the 2.5 km was the optimal resolution allowing at the same time to obtain satisfactory reproducibility of the large scale processes and to explore the urban effects at local scale without being diminished due to a coarse resolution, for a medium size city (even possibly can be considered for a small size city). For other metropolitan areas such as Paris, Rotterdam, St. Petersburg, Shanghai - a similar resolution was chosen, although for Copenhagen (with its flat terrain) the highest possible/ suitable resolution tested was 1.5 km and provided reasonable verification results. Within a selected metropolitan area there could be only a few grid cells having 100% representation of the urban fraction, but taking into account all urban grid cells, the boundaries of the cities (number of cells) could be substantially larger. Moreover, it should be noted that most of existing developed parameterizations in the physics core of any existing NWP model might need a revision when resolutions of 1 km and finer are used.

The Authors provided references to all model components and applications. However, this paper should explicitly provide all 'vital model information' such as vertical structure, horizontal resolutions (with clearly stated limitations), numerical methods and approximations employed in different modules (components), modularity and scalability of components, examples of integration time and computer topology used for benchmarking.

**Response:**
Vertical structure and horizontal resolutions of the model are flexible. Limitations, e.g. due to the hydrostatic approximation, are provided (min 1,5 km for flat terrains, e.g for Copenhagen). Corresponding information, as requested, is included in the revised version.

What is the required computer power, maximum number of computational cores, can the model be run on a heterogeneous architecture with GPUs? All these characteristics should be addressed and tabulated with appropriate references and notes.

**Response**: The model is parallelized with both OpenMP and MPI technics, but it cannot be run on heterogeneous architectures with GPUs. The parallelization algorithm performs 2D decomposition of a modeling domain. The Enviro-HIRLAM can be run on Linux/Unix clusters and CRAY XT5/XC30 high performance computers.
We have not heard of tests where effect on scalability of introducing chemistry, aerosols etc. have been made.
**Changes in manuscript:**
**L427**: The Enviro-HIRLAM modelling domain with horizontal resolution of $0.15^o$ x $0.15^o$ having 310 x 310 grid cells, and 40 vertical hybrid sigma levels extending to pressures less than 10 hPa, covers Europe, North of Sahara, and European Russia. The modeling domain was partitioned into 120 CPU cores and the model was run with time step of 300 seconds.

In several sub-sections, the Authors included a description of earlier versions of the model. Thus, it is not clear to the reader which parameterizations are used in the current version of the Enviro-HIRLAM model. It would be advisable to move these paragraphs to an appendix presenting development stages and perspective of the Enviro-HIRLAM model.

**Response:**
More concrete info about parameterizations used in the considered case studies and in the current version of the Enviro-HIRLAM model is provided in the revised version.

In Section 3 (Modelling system applications), the Authors refer to several earlier publications. It is not clear if the presented manuscript contains any results that were not published. It would be advisable to add a table (in Section 3) with a list of presented experiments and model versions used for simulations together with appropriate references.

**Response:**
Most of results presented in the paper are new (used only in technical reports). We include more accurate references to appropriate papers, if some experiments were considered in previous publications, in the revised version. However, it is difficult to provide such information in a table form.

Also, if a figure is adopted from an earlier publication, a proper reference should be included in a figure caption.

**Response:** Thanks, checked and done.

Pollen module description should be moved from Section 3.3 (Pollen forecast) to Section 2 (system description).

**Response:**
Pollen applications require specific parameterisations of pollen emission sources and other characteristics, so it is more relevant to describe in the section 3.3.

Sub-section 3.4 should be moved and inserted as 3.1

**Response:**
Section 3.1 focuses on the effect of weather while 3.4 is about air-quality forecasting. Although these are two distinct subjects which seem reasonable to address individually.

Overall, the justification of advantages of the on-line approach is not sufficiently demonstrated.

**Response:**
The advantages of the on-line approach were discussed in details in the previous EuMetChem paper (Baklanov et al., 2014).

Verification aspects should be included in a more coherent way. Presented experiments refer to relatively short periods (one summer month). Results for the gas phase chemistry are not discussed.

**Response:**
Yes, we agree that many additional verification and sensitivity experiments are needed for different applications (long-term validation, chemistry mechanisms, etc.). We are working with some of them and they will be in following papers.

The Authors should restructure the manuscript to emphasise the overall modelling philosophy and future directions of the proposed model development and applications.

**Response:**
Thanks. We modified the concluding sections correspondingly in the revised version. However, the overall modelling philosophy and future directions of coupled meteorology-chemistry model development were subjects of our previous papers of EuMetChem, CCMM, etc. (see corresponding references in the paper). Here we focus on the Enviro-HIRLAM model description and its applications.
However, we'd prefer do not change the papers structure dramatically, especially keeping in mind that two other reviewers have found that "The manuscript is well structured and provides a comprehensive presentation of Enviro-HIRLAM development ….".

Specific comments:
The presented comments are in a sequential order and refer to the line numbering in

the presented manuscript.

L22: "Online integrated passive pollutant transport" - the same term should not be used for the simplified approach.

**Response:** Thanks, agree. We mean the online consideration of tracer equations together with other equations at the same time steps (without feedbacks). We modified the sentence.

L27: What is "effective chemistry"?

**Response:** Thanks. Changed to 'cost-efficient'.

L35-36: The section title is too long and awkward.

**Response:** Thanks. The title is shortened.

L68: The style of Figure 1 does not conform to a convention used in scientific publications.

**Response:**
Yes, it might be not the standard/ most common way of the material presentation, but the Figure 1 presents the overall structure of the modelling system, its research development, technical realisation, science education and potential application areas. All these elements are necessary main building blocks in elaboration and maintenance of the modelling system and it is important/useful to present them in such a graphical form.

L108: "current new version" – should be either "current" or "new"?

**Response:** Done.

L128: "main meteorological fields" – please define.

**Response:** It is specified in the text.

L142: How long are the "long-term runs". Please explain and justify.

**Response:** Done: up to one year.

L175-185: The whole section on photolysis rates is confusing and misleading.

**Response:**
For the simplicity of photolysis rates estimation we used the following:
1. For the simple reactions, we estimated the Photolysis rates as a function of number of parameters such as meteorological and chemical inputs including altitude, solar zenith angle, overhead column densities for O3, SO2 and NO2, surface albedo, aerosol optical depth, aerosol single scattering albedo, cloud optical depth and cloud altitude.
2. For the complex reactions, we estimated the Photolysis rates as lookup table using the Tropospheric Ultraviolet-Visible Model (TUV) developed by Madronich and Flocke (1999) and a pseudo-spherical discrete ordinates method (Stamnes et al., 1988) with 8 streams. We run TUV offline and calculated a lookup table of the Photolysis rates, and then we implemented this lookup table under different weather conditions inside our model.

L177: Please explain how the ozone column is set above the model top.

**Response:**
We used the climatological chemical boundary conditions  from MOZART chemical transport model using a monthly average of years 2000–2007 (Horowitz et al., 2003; Emmons et al., 2010). The model

top (50 hPa, corresponding to the lower stratosphere) uses a climatological ozone concentration based on interpolated MOZART ozone fields.
Therefore, the model top layer contains ozone concentrations comparable to the stratosphere. Indeed, we implement the climatological values for computational efficiency during model development and test simulations.

L181: The assertion that the 8-stream method is "the most accurate" should be justified.

**Response:**
The 8-stream method is used and justified in TUV model system, developed by Madronich and Flocke (1999):
Reference: "Madronich, S. and Flocke, S.: The role of solar radiation in atmospheric chemistry; in: Handbook of Environmental Chemistry, edited by: Boule, P., Springer-Verlag, New York, 1–26, 1999"

L282: In Figure 4 X-axes have different units.

**Response:**
Both the left hand plot and the right hand plot in Fig. 4 have x-axes, showing the electromagnetic wavelength. Since the left hand plot shows SW wavelengths, these are given in units of nm, while the LW wavelengths in the right hand plot have units of μm. It is common practice to use these units for SW and LW wavelengths, respectively.

L343: What is "traditional" SL? Please provide a reference.

**Response:** Thanks. We provided the reference to the "traditional semi-Lagrangian" scheme:
Robert, A. 1981. A stable numerical integration scheme for the primitive meteorological equations. Atmos.-Ocean 19, 35–46.

L382: Figure 6: The presented figure alone does not prove that the model can deal with sharp gradients.

**Response:**
Detailed model tests of the ability of ILMC to reproduce sharp gradients are described in Sørensen et al. (2013), in particular Figure 3 and the accompanying discussion in that paper.
The text in the revised version is corrected to avoid confusions.

Line 389: What is TR4?

**Response**: Thanks. It is a mistyping. TR4 should be Eq. (4). Corrected.

Line 390: The mental jump referring to "formal conservation" should be explained.

**Response:**
We have already answered this question to Reviewer 1.
We have added a sentence to clarify that mass-wind inconsistency is a minor problem. The traditional HIRLAM is (at least in principle) wind-mass consistent. In Enviro-HIRLAM, where all moisture fields are transported with the LMCSL scheme, there is no formal consistency, yet, since precipitation is very similar to that in HIRLAM (except for individual convective systems that are chaotic/unpredictable in their nature), the mass-wind inconsistency is small in practice.
A more careful discussion on the issue of mass-wind inconsistence in atmospheric models would require a rather extensive addition. In principle, no monotonic transport schemes can be mass-wind consistent, since the monotonic limiters formally destroy the consistency.
We also add a reference to the paper: Jöckel, P., von Kuhlmann, R., Lawrence, M. G., Steil, B., Brenninkmeijer, C. A. M., Crutzen, P. J., Rasch, P. J., and Eaton, B.: On a fundamental problem in

implementing flux-form advection schemes for tracer transport in 3-dimensional general circulation and chemistry transport models, Q. J. R. Meteorol. Soc., 127, 1035–1052, 2001.

L407: The title is confusing, and the whole section is too long. Half of the first paragraph refers to urban applications, which are discussed in the next section.

**Response:** In the revised version we modified the title to 'Applications for Numerical Weather Prediction' and slightly shortened the text.

L497: It is wrong to assert that higher correlation implies that the model is "closer to observations."

**Response:** We modified the text; the statistical analysis showed that the urban simulation had a reduced bias with respect to observations than the control simulations.

L505: The ability of a weather prediction model (i.e. HIRLAM) to reproduce meso-scale processes at the regional scale should not depend on the use of an urban parameterization. The presented conclusions do not belong in Section 3.2.

**Response:**
Yes, the ability of a weather prediction model to reproduce meso-scale processes does not depend on the use of an urban parameterization. However, since the hydrostaticity of the model was a limitation for increasing the resolution to study the urban impacts, several sensitivity tests demonstrated that the 2.5 km was the optimal resolution allowing at the same time to obtain satisfactory reproducibility of the large scale processes and to explore the urban effects at local scale without being diminished due to a coarse resolution (as fraction of urban areas in grid cells of coarser resolution became very diluted).

L654: The calculations were analysed for one month (July 2010) only. Thus, the sentence is too general.

**Response:** Thanks. The sentence is changed in the revised version.

L656: "crude model resolution" – what does it mean?
The use of the English language:
The Authors should pay particular attention to the use of articles, prepositions and tenses in the revised manuscript. Also, the Authors used words that do not exist i.e.
Line 255: 'to split' is an irregular verb – the simple past tense is 'split', or words in a wrong context i.e. Line 187 'Heterogenic chemistry' should be 'Heterogeneous chemistry'.

**Response:** Thanks. We checked and corrected the language in the revised version.

Recommendation:
In the opinion of this reviewer, the presented manuscript could constitute an important contribution documenting the Enviro-HILRAM model. The paper should be published after major revisions.

**Response:** Thanks a lot. We do our best for that.

---

## Author Comment (AC4) · 31 Mar 2017

**Authors response to comments of the Referee #3**

**We thank the Anonymous Referee #3 for the interesting and important comments on our manuscript. All the individual comments are addressed below in red.**

The manuscript presents the online coupled model Eviro-HIRLAM, which is well known in the atmospheric modelers community. The manuscript is well structured and provides a comprehensive presentation of Enviro-HIRLAM development with a description of the different approaches and physical schemes implemented during the model evolution. The computational schemes and parameterizations adopted by the models are properly introduced and referenced. A minor shortcoming of this approach is that it is somewhere not very clear to the reader which computational scheme is the one chosen for the present version of the model or what alternatives are provided to the user.

**Response**:
The revised version provides more information about computational schemes chosen for the present version of the model.
The LMCSL with monotonic filter is the scheme chosen for the present version. As of now one may be in doubt if this just an option.

A relevant number of applications are referenced for almost all the model development fields. Some of the items (e.g. pollen) are described providing explicit summary of the overall results that make the paper more readable and useful for a reader that is not willing to read the large number of referenced papers and documents. Other examples of application are mainly discussed through references and do not allow the reader to appreciate the model effectiveness and the improvement offered by the online modelling approach.

**Response**:
We agree with the reviewer: the pollen part was not described in previous publications, so we did it in more details. Other aspects, considered in specific previous papers, are only briefly described here with corresponding references.

If the general approach of online coupling is physically sound and it can be agreed that it will probably become the prevailing modelling approach in the next future, the manuscript does not clarify, through its application examples, to what extent the online coupling and the main parameterizations introduced (e.g. urbanization) provide an improvement of model capability to predict observed pollutant concentrations and key meteorological parameters. An improvement of the analysis of the online coupling effectiveness is desirable and would make the manuscript more complete, interesting and valuable.

**Response**:
These issues are really very important, but the previous EuMetChem paper (Baklanov et al., 2014) considered them more comprehensive and not only for the Enviro-HIRLAM model.

Text and figures include a large number of acronyms for project names, parameterization schemes, etc. Even if many of them are known, it is quite difficult for the reader to know and remind all their meaning. It would be helpful to add an acronym legend section.

**Response**: Thanks. Done.

Specific comments:
Section 1. Methodology
Lines 72-75 The authors say that Enviro-HIRLAM is being used for different research
project, but most cited project have already concluded they activity. In the Figure 1
lowest box most project mentioned as ongoing are finished since a few years.

**Response**:
Many previous and recent projects are mentioned in the text (FUMAPEX, MEGAPOLI, MACC,
PEGASOS, MarcoPolo, EuMetChem, CarboNord, CRAICC-PEEX, CRUCIAL, …).
We have adjusted the info in the Figure 1 lowest box correspondingly.

Section 2.1 Modelling system structure
Line 92 The URL http://hirlam.org/trac/wiki/ is password protected and therefore not
accessible to the reader. It should be substituted with an open access web site.

**Response**:
This is the policy of the HIRLAM consortium. We are in contact with the HIRLAM web-master to
open this link or to provide another open one.

Section 2.3 Atmospheric chemistry
It is not clear if the "tropospheric sulfur cycle" is a simple scheme alternative to the
CBM-Z, that is presently maintained for simplified simulations (what is the specific interest?),
or if it is an obsolete option which is going to be abandoned. It is not specified
how the CBM-Z gas-phase chemistry scheme is interfaced with the M7 aerosol module.
Due to the relevance of secondary particle production modelling, more details
would be appreciable to provide a comprehensive model description.

**Response**:
The tropospheric sulfur cycle chemistry is used together with M7 aerosol microphysics module
because of its relative simplicity and low computational cost. The CBM-Z gas-phase chemistry is not
interfaced with the M7 aerosol module because of several reasons: 1) the aerosol microphysics
module does not include Secondary Organic Aerosols, therefore, there is no need of complex gas-
phase mechanism with Volatile Organic Compounds related reactions and 2) it is too computationally
expensive to use CBM-Z together with M7 for both weather and atmospheric composition prediction.

Lines 171-172 The authors say they "use KPP tools to create the gas-phase chemical
mechanisms including the solvers for three chemical mechanisms." What are the three
mentioned chemical mechanisms? Only two of them have been previously presented:
a) Tropospheric Sulfur Cycle, b) Gas-phase chemistry (CBM-Z).

**Response**:
Indeed, during the validation stages of creating the gas-phase schemes we used the Kinetic
Preprocessor (KPP) (Sandu et al., 2006); we used KPP to create the Fortran code of three gas-phase
schemes CBM-Z (Zaveri et al. 1999), GEOS-CHEM (Evans et al. 2003) and the Regional
Atmospheric Chemistry Model "RACM" ( Stockwell et al,1997).
For the chemical weather predication propose, GEOS-CHEM and RACM are very computational
expensive schemes. GEOS-CHEM and RACM schemes include a large number of chemical reactions.
For more simplicity we cooperate with Dr. Rahul Zaveri (Personal communication with Dr. Ashraf
Zakey) in order to simplify CBM-Z and online coupled it with the Enviro-HIRLAM Model.
"Tropospheric Sulfur Cycle scheme" is a very simple sulphur scheme (Easter et al., 2004). It was
ported from HAM without use of the KPP tool. Reference: Easter, R. C., S. J. Ghan, Y. Zhang, R. D.
Saylor, E. G. Chapman, N. S. Laulainen, H. Abdul-Razzak, L. R. Leung, X. Bian, and R. A. Zaveri

(2004), MIRAGE: Model description and evaluation of aerosols and trace gases, J. Geophys. Res.,109, D20210,doi:10.1029/2004JD004571"

Lines 172-173 The authors say that Rosenbrock solver is usually selected. Why?

**Response**:
The Rosenbrock solver is mostly used within the air quality models communities because it is computational fast.

Line 190 What is "NWP-Chem-Liquid"?

**Response**: The "NWP-Chem-Liquid" is a thermodynamic equilibrium model, described in Korsholm et al. (2008). Many gas-phase species are water soluble and sulphate and ammonia together with water take part in binary/ternary nucleation. In order to consider these processes, a simplified liquid-phase equilibrium mechanism with the most basic equilibria is included in NWP-Chem-Liquid. This equilibrium module is solved using the analytical equilibrium iteration method (Jacobson, 1999).

Section 2.4. Aerosol formation, dynamics and deposition
Line 197 Is CAC still available in Enviro-HIRLAM or it is mentioned only for historical development reasons?

**Response**:
No, it is not used in the last reference version and in the described simulations, but can be called for specific studies. See e.g. Gross and Baklanov (2004), Korsholm (2009).

Lines 205-206 Is the aerosol type identity maintained through the model simulation and provided as separated output contribution to the total PM?

**Response**: Different aerosol types mentioned in the model description and simulations (as described in section 2.4) are provided as separate species in the model outputs along with lumped $PM_{10}$ and $PM_{2.5}$.

Section 2.5. Emission modules and pre-processor
Line 254 Does wildfires emission module consider PM only or gas phase pollutants too?

**Response**:
The wildfires emissions were from the Finish Meteorological Institute - Fire Assimilation System v.1.1, which provides total lumped emissions. The total was split according to Andreae and Merlet, 2001 in organic and black carbon, and gaseous emissions of $SO_2$ only. The gas-phase pollutants like Nitrogen Oxide (NO) and Volatile Organic Compounds (VOCs) were not considered or processed.

Line 274 What are "transported modes"?

**Response**:
The "transported" mineral size mode in the GADS/OPAC data set (Köpke et al. 1997) is usual aerosol size mode that comes in addition to the more standard "nucleation", "accumulation" and "coarse" mineral size modes. Köpke et al. (1997) uses the "transported" size mode to describe aerosols that have been transported over a long distance, for instance Saharan aerosols that have been blown to the Atlantic ocean.

Section 2.7. Urban parameterizations and models urbanization
This section is relevant because it highlights the need for a mass conserving transport scheme in on-line coupled NWP and ACT models. For offline coupling this request is less strict because mass consistency is usually guaranteed by the coupler module.

Line 311 Bracket missing.

**Response**: Thanks. Done.

Line 312 Grid nesting is an effective technique to increase model resolution but it is rather confusing to consider it a method to represent urban areas.

**Response**:
The nesting technics and downscaling methods are actively and successfully used for urban areas to reach the necessary resolution for resolving or parameterisation of urban features and effects. The details of this approach was described e.g. in Baklanov and Nuterman (2010).
With respect to metropolitan areas, the downscaling for finer/ better resolution allows to reproduce smaller scale meteorological patterns, and then these patterns are further modified through running the urban modules such as BEP, SM2U, BEM, etc. only for grid cells where the cities are presented.
The text of this section is modified correspondingly.

Line 315 The "calculation of the urban mixing height based on prognostic approaches" is neither described nor commented in the following text.

**Response**:
Thanks. This issue was published previously in BLM papers Zilitinkevich et al. (2002) and Zilitinkevich and Baklanov (2002). Some clarifications were done: additional text and references on specific papers are included.
References:
Zilitinkevich, S. and A. Baklanov, 2002: Calculation of the height of stable boundary layers in practical applications. Boundary-Layer Meteorology, 105(3), pp. 389-409.
Zilitinkevich, S., A. Baklanov, J. Rost, A.-S. Smedman, V. Lykosov & P. Calanca, 2002: Diagnostic and prognostic equations for the depth of the stably stratified Ekman boundary layer. Quarterly Journal of the Royal Meteorological Society, 128, pp. 25-46.

Section 2.8. Transport schemes
Line 371 Is hat symbol missing on "modified weight" in equation 6 ?

**Response**: Thanks. Yes, it is. There should be a hat over the W. Corrected in the revised version.

Line 377 ""is are" should be corrected

**Response**: Thanks. Done.

Lines 388-390 This sentence concerning Enviro-HIRLAM mass consistency for tracer transport should be better explained and discussed. What are the possible limitations caused by this lack of mass conservation? What is TR4?

**Response**: Thanks. It is a mistyping. TR4 should be Eq. (4).
We have already answered this question to Reviewer 1.
We have added a sentence to clarify that mass-wind inconsistency is a minor problem. The traditional HIRLAM is (at least in principle) wind-mass consistent. In Enviro-HIRLAM where all moisture fields are transported with the LMCSL scheme there is no formal consistency, yet, since precipitation is very similar to that in HIRLAM (except for individual convective systems that are chaotic/unpredictable in their nature), the mass-wind inconsistency is small in practice.
A more careful discussion on the issue of mass-wind inconsistence in atmospheric models would require a rather extensive addition. In principle no monotonic transport schemes can be mass-wind consistent since the monotonic limiters formally destroy the consistency.
We also add a reference to the paper: Jöckel, P., von Kuhlmann, R., Lawrence, M. G., Steil, B., Brenninkmeijer, C. A. M., Crutzen, P. J., Rasch, P. J., and Eaton, B.: On a fundamental problem in

implementing flux-form advection schemes for tracer transport in 3-dimensional general circulation and chemistry transport models, Q. J. R. Meteorol. Soc., 127, 1035–1052, 2001.

Section 3 Modelling system applications
What are the mentioned "EnvCLIMA, Enviro-HIRHAM"?

**Response**: Thanks. It is clarified/modified in the revised version.

Lines 415-418 Do the mentioned temperature changes due to indirect effects improve model results? How relevant is the improvement? The reference given by the authors is to a Project report that can be hardly available, not to a journal publication. In the following sentence (lines 420-421) the authors mention a marginal improvement on surface temperature. They also mention a redistribution effect on NO2 concentration, but they do not specify is this effect improves model results.

**Response**:
Yes, these study results were described only in reports and proceeding papers. Corresponding journal paper is under preparation. The improvements due to the indirect effects exist (as shown e.g. in Fig 9), but the existing parameterisations of indirect effects need further improvement and evaluation. Several publications of different authors (e.g. Vogel et al., 2015) also stressed that these indirect mechanisms are the most uncertain and need further improvements.
We have answered in more details on the similar question to the Reviewer 1.

Lines 442-444 and Figure 9 The authors say "the ENV run bias for precipitation with respect to its frequency and amount has been decreased compared to the REF model run (Fig. 9)." Legends printed on the pictures seem opposite to what indicated in the caption (Enviro-HIRLAM on the left). Results showed in Figure 9 seem different during different parts of simulation: until July 21st the right side simulation seems better, while the left side one seems better during the last part of the simulation. What is the difference of the overall biases?

**Response**:
It is an unfortunate mistake; the left and the right figures must be swapped.
According to observations at WMO station 6670 at Zurich, Switzerland, the mean 12 hours accumulated precipitation in July 2010 was 0.97 mm, the median was 0 mm and the precipitation variance at the site was 7.52. As for the reference HIRLAM run, the modeled monthly mean, the median and the variance of 12 hours accumulated precipitation are equal to 1.83 mm, 0.14 mm and 16.90, respectively. The Enviro-HIRLAM model with aerosol-cloud interactions predicted the mean value of 1.16 mm, the median of 0 mm and the variance of 9.53 of 12 hours accumulated precipitation for the same month. That means the reference model tends to overpredict both the precipitation frequency and its amount, but the aerosol-cloud feedbacks in the Enviro-HIRLAM model reduce such over-prediction tendencies. Moreover, the values of Fractional Bias of Ref-HIRLAM (-0.61) and Enviro-HIRLAM (-0.18) along with Normal Mean Square Error values of Ref-HIRLAM (4.17) and Enviro-HIRLAM (3.45) show improvement of the Enviro-HIRLAM prediction score comparing to Ref-HIRLAM.

Lines 480-489 A grid size of 2.5 km seems quite crude to resolve Bilbao city. In x and y directions the city seems to be described by 2 to 4 grid cells which can be hardly considered sufficient to develop a "urban signal". Why has not been used a finer resolution? Is it due to the hydrostatic model limitations?

**Response:**
Yes, the hydrostatic approximation of the model was a limitation to increase the resolution to perform the urban simulations. However, sensitivity tests demonstrated that the 2.5 km was the optimal resolution allowing at the same time to obtain satisfactory reproducibility of the large scale processes

and to explore the urban effects at local scale without being diminished due to a coarse resolution, for a medium size city (even possibly can be considered for a small size city). For other metropolitan areas such as Paris, Rotterdam, St. Petersburg, Shanghai - a similar resolution was chosen, although for Copenhagen (with a flat terrain) the highest possible/ suitable resolution tested was 1.5 km and provided reasonable verification results. Within a selected metropolitan area there could be only a few grid cells having 100% representation of the urban fraction, but taking into account all urban grid cells, the boundaries of the cities (number of cells) could be substantially larger. Moreover, it should be noted that most of existing developed parameterizations in the physics core of any existing NWP model might be also needed to be revised when resolutions of 1 km and finer are used.

Figure 10 Why different land use classifications have been used for the two considered cities? What is the P01 modelling domain mentioned in the caption?

**Response:**
Depending on a country-by-country basis and national architectural specifics, different metropolitan areas could have different types of urban fabric with specific aerodynamical and morphological characteristics of urban districts. The size of the Bilbao metropolitan area is at least 10 times less than the Paris metropolitan area. Therefore, to harmonize the urban classification we considered that Bilbao had a Residential high and low density districts (RLD, RHD, respectively); while Paris metropolitan areas was characterised by a residential district (RD) and the city centre (CC). Also, note that for the land-use classification of the Bilbao metropolitan area, a local land-use database was used and for Paris, the land-use database CORINE 2000 was applied. (Gonzalez-Aparicio et al. 2010). The P01 domain is just one of names for the modelling domains created for the Enviro-HIRLAM model runs with the focus on the Paris metropolitan area located in the centre of the domain. It has been removed from the caption.

Line 498 Does 10% improvement refer to the correlation value?

**Response:** Yes, it is referred to the overall correlation values.

Lines 499-500 It is not clear how the mentioned correlations have been computed. Time correlation for separated hours? How many stations have been used to compute the mentioned correlations?

**Response:**
The correlations were computed for the winter and summer months, simulated averaged over each hour of the day (e.g. considering the diurnal cycle), at each of the three types of locations considered (urban, suburban and rural – Figure 11a).

Lines 501-504 Where the mentioned results for Bilbao better than those obtained without urbanization? Was the improvement significant?

**Response:**
The results have been mentioned in Gonzalez-Aparicio et al. (2013). The results of those simulations were significant since we showed that the Enviro-HIRLAM model (urbanized version) was able to simulate the effect of the Urban Heat Island over a medium size city located in a coastal and complex terrain area characterized by land-sea breeze.

Lines 512-535 The authors show that model urbanization allows to describe UHI phenomenology in Paris and Bilbao, but they do not discuss if the urbanization improves results and reduces possible model bias with respect to urban observations.

**Response:**
Gonzalez-Aparicio et al. (2013) discussed the urban parameterization implementation in the Enviro-HIRLAM model and the improvement with respect to the control simulations for the Bilbao city. The

urban effect and the results were compared with the results obtained in an experimental campaign over the city.

Lines 635-639 The mentioned effects of aerosol feedbacks on chemical composition are quite interesting. Did the mentioned changes on NO2 and O3 improve model results and increase its capability to reproduce measured values?

**Response**:
Unfortunately it was just a sensitivity study and a proper long-term validation was not realised yet. So, we prefer to avoid conclusions.

Figure 15 Right side color scale legend needs correction. How are correlations for separated hours computed?

**Response**:
We do not know how to change the legend scale, because the referee did not specify any required correction.
In order to compute correlation coefficients on diurnal cycle, the Enviro-HIRLAM model output was collected for separate time slices (00, 03, 06, … 21 UTC) and observation sites, and then the correlation coefficients were computed separately for each time-slice and site.

Lines 675-677 The authors mention new model applications without providing any detail about recent results potentially relevant and interesting for the readers. The mentioned feedback mechanisms evaluation is one of the key point of the paper.

**Response**:
Unfortunately we cannot answer on all the questions of online chemistry-meteorology modelling in one this paper. Some potential applications are just briefly mentioned in the paper and they are topics for further studies and analysis. In particular the results of the CarboNord project for Black Carbon feedbacks for the Arctic are now under analyses and will be published in a separate paper.

Section 4 Conclusions
Lines 692-702 These sentences contain repetitions of the same concepts that could be removed.

**Response**: Thanks. Done.

---

## Author Response (AR3)

Dear Dr. Jason Williams,

We fully agree with your comments about the Sections 4 and 5.
They are rewritten/restructured following your requirements.
In particular:
- the first paragraph of the further discussion section merged into the conclusions,
- discussions based on previous studies/findings moved to the Further discussion section,
- the text of sections 4 and 5 modified correspondingly (the last changes are visible below).

Small changes are also made in the Acknowledgments section.

Thanks a lot for your very useful comments.
Best regards,
Alexander

**Topical Editor Decision: Publish subject to minor revisions (Editor review)** (30 May 2017) by Jason Williams
Comments to the Author:
Dear authors,

I have read your revised submission and feel that the document is now more in line with the guidelines for authors related to GMD submission.
There is still one outstanding issue with the conclusions section related to how the content is spilt between the new Further Discussion and Conclusions sections. The conclusions should read as an entirely independent summary of the entire paper, thus abbreviations defined in the manuscript cannot be used. Therefore as a final action, please merge the first paragraph of the further discussion section into the conclusions. I am looking for a lead sentence such as e.g. "In this manuscript we have provided a comprehensive description of the Enviro-HIRLAM model ... ", currently found in the Further Discussion section. One important thing is that you cannot use other peoples findings in your own conclusions unless your have conducted similar independent simulations only available in this manuscript. Therefore, previous findings that chemistry is/isn't important for applications should be moved to the Further discussion section if based on the findings of previous studies. In the conclusions section you can add something about previous applications e.g. "
[revised manuscript text omitted]